# Fabrication and Characterization of Tedizolid Phosphate Nanocrystals for Topical Ocular Application: Improved Solubilization and In Vitro Drug Release

**DOI:** 10.3390/pharmaceutics14071328

**Published:** 2022-06-23

**Authors:** Mohd Abul Kalam, Muzaffar Iqbal, Abdullah Alshememry, Musaed Alkholief, Aws Alshamsan

**Affiliations:** 1Nanobiotechnology Unit, Department of Pharmaceutics, College of Pharmacy, King Saud University, Riyadh 11451, Saudi Arabia; makalam@ksu.edu.sa (M.A.K.); aalshememry@ksu.edu.sa (A.A.); malkholief@ksu.edu.sa (M.A.); 2Department of Pharmaceutics, College of Pharmacy, King Saud University, Riyadh 11451, Saudi Arabia; 3Department of Pharmaceutical Chemistry, College of Pharmacy, King Saud University, Riyadh 11451, Saudi Arabia; muziqbal@ksu.edu.sa; 4Bioavailability Unit, Central Lab, College of Pharmacy, King Saud University, Riyadh 11451, Saudi Arabia

**Keywords:** tedizolid phosphate, nanocrystals, thermal characterization, in vitro release, stability

## Abstract

Positively charged NCs of TZP (0.1%, *w*/*v*) for ocular use were prepared by the antisolvent precipitation method. TZP is a novel 5-Hydroxymethyl-Oxazolidinone class of antibiotic and is effective against many drug-resistant bacterial infections. Even the phosphate salt of this drug is poorly soluble, therefore the NCs were prepared for its better solubility and ocular availability. P188 was found better stabilizer than PVA for TZP-NCs. Characterization of the NCs including the particle-size, PDI, and ZP by Zeta-sizer, while morphology by SEM indicated that the preparation technique was successful to get the optimal sized (151.6 nm) TZP-NCs with good crystalline morphology. Mannitol (1%, *w*/*v*) prevented the crystal growth and provided good stabilization to NC_1_ during freeze-drying. FTIR spectroscopy confirmed the nano-crystallization did not alter the basic molecular structure of TZP. DSC and XRD studies indicated the reduced crystallinity of TZP-NC_1_, which potentiated its solubility. An increased solubility of TZP-NC_1_ (25.9 µgmL^−1^) as compared to pure TZP (18.4 µgmL^−1^) in STF with SLS. Addition of stearylamine (0.2%, *w*/*v*) and BKC (0.01%, *w*/*v*) have provided cationic (+29.4 mV) TZP-NCs. Redispersion of freeze-dried NCs in dextrose (5%, *w*/*v*) resulted in a clear transparent aqueous suspension of NC_1_ with osmolarity (298 mOsm·L^−1^) and viscosity (21.1 cps at 35 °C). Mannitol (cryoprotectant) during freeze-drying could also provide isotonicity to the nano-suspension at redispersion in dextrose solution. In vitro release in STF with SLS has shown relatively higher (78.8%) release of TZP from NC_1_ as compared to the conventional TZP-AqS (43.4%) at 12 h. TZP-NC_1_ was physically and chemically stable at three temperatures for 180 days. The above findings suggested that TZP-NC_1_ would be a promising alternative for ocular delivery of TZP with relatively improved performance.

## 1. Introduction

Due to the unique anatomy and physiology of eyes, the development of ocular drug delivery (ODD) is a major challenge to the formulators [1,2,3,4]. The aim of ODD is to maximize drug concentration in the ocular sites to achieve maximum beneficial effects at low dosing [5,6]. Some conventional formulations for ODD including the eye drops (solutions, suspensions, emulsions, etc.) containing poorly soluble or low permeable drugs have shown low ocular availability (approximately 1–5% of the applied dose) of the drugs [7,8]. Due to normal physiology of eyes (blinking, precorneal loss, nasolacrimal drainage), the conventional formulations are not fully successful to attain effective therapeutic drug concentrations in the eyes [9,10]. Application of ointments, to some extent, could resolve the poor ocular bioavailability issues though they cause blurring of vision [11]. Due to the unique anatomical structure and natural defense mechanism of eyes, the permeation of applied drugs is hindered and leads to poor ocular bioavailability [8,12,13].

Nanotechnology-based ODD approaches have successfully overcome the problems associated with the conventional dosage forms and enhance the ocular bioavailability of poorly soluble drugs [14]. The technologies including the supercritical assisted (CO_2_-mediated) method to produce the nano-formulations (e.g., liposomes) [2,15,16,17], nanocrystals by top-down or bottom-up techniques [5,7,9,15,18], polymeric micelles [19], polymeric nanoparticles (NPs) [3,20], nanoemulsion [1,21], microemulsion [22], solid lipid nanoparticles [10,23,24], niosomes [25,26], dendrimer nanoparticles [27,28] etc.

Despite having some drawbacks, the nanocarriers in ODD have been explored well. The perspective of nanocrystals (NCs) for ocular use has been comparatively unnoticed because of the availability of numerous established bioadhesive polymeric NPs [8,29]. Therefore, we developed the NCs of tedizolid phosphate (BCS Class-IV) for topical ocular application. The potential application of NCs for oral and parenteral delivery has already been carried-out [9,30,31,32,33]. It also shows substantial possibilities for ODD [7]. The application of NCs has not been well explored for ocular applications, hence it opened new opportunities for the cure of ocular diseases [4,9,34,35].

The NCs can be a successful alternative to the pharmaceutical industries as they have achieved commercial success to some extent in oral, parenteral, and ODD of highly lipophilic, poorly soluble, and low permeable drugs [30,36]. We believe that in near future the NCs would support the development of robust and effective therapies for ocular disorders. The NCs-based formulations have received prime interest as a feasible and commercial substitute as evidenced by some product patents based on NCs of poorly soluble drugs [37,38].

The NCs-suspension of poorly soluble drugs such as dexamethasone, hydrocortisone and prednisolone have improved the ocular bioavailability as compared to their micro-crystalline suspensions and aqueous solutions [9,39]. The NCs as ODDS can increase ocular retention, transcorneal permeation and hence bioavailability [7,9]. The positively charged formulations may interact electrostatically with the ocular mucin layer which can prolong the retention of the formulation that facilitates the increased transcorneal permeation and improve the ocular bioavailability of the drug [8]. Also, the high absolute zeta-potentials provide potential stability to any colloidal systems. 

Above finding encouraged us to develop the NCs for topical ocular application of a highly lipophilic and poorly aqueous soluble drug (TZP). The TZP is a novel 5-Hydroxymethyl-Oxazolidinone class of antibiotic [40] having excellent antibacterial activity against *Mycobacterium tuberculosis* [41], methicillin-resistant *Staphylococcus aureus* (MRSA) and many other Gram-positive bacteria [42] including their infections in the eyes [8]. TZP is a prodrug and converted into tedizolid (TDZ) in vivo by alkaline and acid phosphatases, hence either TZP or TDZ can be chosen for its ocular uses [43]. The chemical structure of TDZ is represented in Figure 1.

The prodrug TDZ was chosen as its phosphate salt has better aqueous solubility [40] and hence, would result in an improved ocular availability of the active moiety. Also, the presence of modified side chains at the C5 site of the oxazolidinone nucleus of TDZ provides an excellent activity against vancomycin-resistant *enterococci* [44] and some linezolid-resistant strains [45].

The antisolvent precipitation method was adopted to develop the NCs of TZP [7]. The PVA and P188 were tried as stabilizers to optimize the NCs. The developed TZP-NCs were characterized and evaluated for their suitability and potential as nanosystems for the ocular delivery of TZP.

## 2. Materials and Methods

### 2.1. Materials

Tedizolid base (CAS: 856866-72-3) and tedizolid phosphate (C_17_H_15_FN_6_O_6_P; MW 450.32 Da) (CAS: 856867-55-5) with more than 98% purity, were purchased from “Beijing Mesochem Technology Co. Ltd. (Beijing, China)”. Tween-80 (CAS: 9005-65-6) and HPLC grade methanol (CAS: 67-56-1) were purchased from “BDH Ltd. (Poole, England)” Polyvinyl alcohol (Mw 16,000) (CAS: 9002-89-5), Poloxamer-188 (Pluronic-F68) (CAS: 9003-11-6), Chloroform (CAS: 67-66-3), Ethanol (CAS: 64-17-5), Dimethyl Sulfoxide (CAS: 67-68-5), Polyvinylpyrrolidone-K30 (CAS: 9003-39-8), Vit-E-TPGS (CAS: 9002-96-4), Sodium Lauryl Sulfate (CAS: 151-21-3), and Benzalkonium chloride (CAS:63449-41-2) were purchased from Sigma Aldrich, St. Louis, MO, USA. Stearylamine (CAS: 124-30-1) was purchased from Alpha Chemika, Mumbai, India. Mannitol (CAS: 69-65-8) was purchased from Qualikems Fine Chem Pvt. Ltd. (Vadodara, India). Purified water was obtained using a Milli-Q^®^ water purifier (Millipore, Molsheim, France). All other chemicals and solvents were of analytical grade and HPLC grade, respectively.

### 2.2. Methods

#### 2.2.1. Chromatographic Analysis of TZP

A reverse phase (RP) high-performance liquid chromatography (HPLC) with UV-detection (at 251 nm wavelength) was used for the quantification of TZP after slight modification of the reported HPLC method [46]. In brief, the HPLC system (Waters^®^ 1500-series controller, Milford, MA, USA) was used, which was equipped with a UV-detector (Waters^®^ 2489, dual absorbance detector, Milford, MA, USA), a binary pump (Waters^®^ 1525, Milford, MA, USA), automated sampling system (Waters^®^ 2707 Autosampler, Milford, MA, USA). The HPLC system was monitored by “Breeze software”. The TZP was analyzed by injecting 30 µL of the supernatant into the analytical column. An RP, C_18_ column (Macherey-Nagel 250 × 4.6 mm, 5 μm) at 40 °C was used. The mobile phase consisted of 65:35 *v*/*v* of 0.02 M Sodium acetate buffer (the pH was adjusted to 3.5 by Hydrochloric acid) and acetonitrile was pumped isocratically at 1 mL/min of flow rate. The total run time was 10 min. The standard stock solution of TZP was prepared in methanol (100 μgmL^−1^) and working standard solutions (0.25–50 μgmL^−1^) were prepared by serial dilution of the stock solution with 65:35, *v*/*v* mixture of mobile phase. The results of the method were briefly explained in Appendix A.

#### 2.2.2. Preparation of Nanocrystals

The antisolvent precipitation method (bottom-up technique) was used to prepare the nanocrystals (NCs) of TZP [31,47]. Accurately weighed 10 mg of TZP was dissolved in 1: 1, *v*/*v*, mixture of chloroform and ethanol with 0.2 mL of Dimethyl Sulfoxide (DMSO). This solution (2 mL) was added drop-wise (1.5 mL/min) into 20 mL of aqueous phase containing varying concentrations (0.5 to 2.5%, *w*/*v*) of stabilizers (P188 and PVA) at continuous magnetic stirring (800 rpm) to obtain a homogenous suspension. The mixture was homogenized (IKA^®^-WERKE, GMBH & Co., Staufen, Germany) for 5 min at 21,000 rpm, followed by pulsative probe sonication (Sonics & Materials, Inc., Newtown, CT, USA) at 40% power for 60 s (10 s each cycle) on ice bath. The probe sonication during this process provided added stability to the TZP-NCs by minimizing the metastable zone for TZP and its super-saturation level. Organic to aqueous phase ratio was 1:9, *v*/*v*. The organic solvents were then evaporated by continuous magnetic stirring (12 h) at room temperature to get the suspension of NCs. The suspended NCs were purified by washing (in triplicate) with Milli-Q^®^ water to get rid of the excess stabilizers, and collected using ultracentrifugation (Preparative ultracentrifuge, WX-series by Hitachi Koki, Ibaraki, Japan) at 4 °C for 20 min and at 30,000 rpm. The obtained pellets of NCs were then resuspended in 10 mL of Milli-Q^®^ water containing different concentrations of mannitol (0, 1, 2.5 and 5%, *w*/*v*) as protectant during freeze-drying (FreeZone-4.5 Freeze Dry System, Labconco Corporation, Kansas, MO, USA). The freeze-dried and free flowing product was then stored at −20 °C for further characterization.

#### 2.2.3. Measurement of Average Size, Polydispersity-Index and Zeta-Potential

The particle size (hydrodynamic diameter) and polydispersity-index (PDI) of the NCs were measured by Differential Light Scattering (DLS) technique using Zetasizer Nanoseries, (Nano-ZS, Malvern Instruments, Worcestershire, UK). The results are the mean particle size, which is the intensity versus average diameter/thickness of the NCs majority population, and PDI is the measure of the size distribution width. The suspension of NCs was diluted with Milli-Q^®^ water to get their appropriate concentration and the mean values were calculated from three measurements. The zeta-potential (ZP) of the NCs was measured by the same instrument at ambient temperature in the original dispersion medium of the NCs (aqueous solution of stabilizer with and without mannitol). The installed software (DTS Version 4.1, Malvern, Worcestershire, UK) with this instrument automatically measured the electrophoretic mobility of the NCs and transformed it to zeta potential using the “Helmholtz-Smoluchowski equation”.

#### 2.2.4. Scanning Electron Microscopy (SEM)

The morphological characteristics of the NCs were visually observed by micrographs of the samples obtained through SEM (Zeiss EVO LS10, Cambridge, UK) imaging following the gold-sputter technique. In this technique, the dried samples were coated with gold in the “Q-150R Sputter Unit” from “Quorum Technologies Ltd. (East Sus-sex, UK)” in Argon atmosphere at 20 mA for 1 min. Micrographs were at the accelerating voltage of 5 kV and 20–60 KX of magnification.

### 2.3. Differential Scanning Calorimetry (DSC)

Thermal analysis by DSC was carried out on the pure drug (TZP), Poloxamer-188 (P188), mannitol, physical mixture of TZP, P188, and mannitol (PM) and freeze-dried NC_1_ using DSC-8000 (Perkin Elmer Instruments, Shelton, CT, USA) at 10 °C·min^−1^ scan rate. Approximately 2.5–5 mg of weighed samples were hermetically sealed in aluminum pans for this purpose. The DSC was calibrated with pure Indium (having a 156.60 °C melting point and 6.80 cal·g^−1^ heat of fusion). The DSC curves were obtained at 10 °C·min^−1^ scanning rate over 40–280 °C temperature range under the inert atmosphere purged with N_2_ at 20 mL·min^−1^ of flow rate. The results from the obtained curves were further analyzed by the installed software Pyris V-11.

### 2.4. Fourier Transform Infrared Spectroscopy (FTIR)

The FTIR spectra of pure drug (TZP), Poloxamer-188 (P188), mannitol, physical mixture of TZP, P188, and mannitol (PM) and freeze-dried NC_1_ using a BRUKER Optik GmBH (Model ALPHA, Ettlingen, Germany) attached with the software OPUS Version-7.8. The analytes were triturated Potassium bromide (KBr) in 1:100 (*w*/*w*) ratio using mortar and pestle; thereafter, the mixture was compressed to pellets by hydraulic press. The spectra of the pellets were obtained from 4000 to 500 cm^−1^ wavenumber using 2 cm^−1^ resolutions.

### 2.5. Powdered X-ray Diffraction (PXRD)

The powder X-ray diffractions of the above-mentioned samples were carried out by Ultima-IV Diffractometer (Rigaku, Inc., Tokyo, Japan) over the 2*θ*(°) range from 3 to 60 at 0.5 degree/min of scan rate to examine the crystallinity of the samples. The X-ray tube (anode material) was Cu with Ka2 elimination of 0.154 nm, monochromatized with the graphite crystal. The diffraction pattern was obtained at 40 mA of tube current and 40 kV of tube voltage for the generator with step scan mode (step size 0.02° and counting time was 1 s per step).

Furthermore, the crystallite size of NC_1_ as compared to TZP-pure was determined using the data obtained by XRD analysis [48] following the Scherrer Equation (1):(1)$$D=\frac{K\lambda }{\beta cos\theta }$$
where *D* = average size of crystallite size, *K* = Scherrer constant “(i.e., 0.68 to 2.08, 0.94 for spherical crystallites with cubic symmetry)”, *λ* = wavelength of X-ray, *CuKa2* = 1.5406 Å, *β* = the line broadening at FWHM in radians which describes the transmission characteristics of an optical band-pass filter and *θ* = Bragg’s angle.

### 2.6. Physicochemical Characterization of Suspension of NCs

The equivalent amount of the freeze-dried NC_1_ (freeze-dried with 1%, *w*/*v* mannitol) was suspended in 5%, *w*/*v* dextrose solution to get a final TZP concentration (0.1%, *w*/*v*). Benzalkonium chloride (BKC, 0.01%, *w*/*v*) was added to the suspension as preservative [49]. The clarity/transparency of the NC_1_ suspension was examined visually under normal light against dark and white. The pH was checked at room temperature and steady state using the pH meter (Mettler Toledo MP-220, Greifensee, Switzerland), which was previously calibrated with standard buffer solutions of pH 4.0, 7.0, and 10.0. Osmolarity, that is the measure of solute concentration (the number of Osmoles of solute/L of a solution (Osml·L^−1^), was measured by Osmometer (Fiske Associate Inc., Waterford, Pennsylvania, USA) and viscosity of the NC_1_ suspension was determined by “Sine-wave VIBRO VISCOMETER, Model SV-10, having range 0.3~10,000 mPa·s or cP, A & D Co. Ltd., Tokyo, Japan)” by the Tuning-fork vibration method. In this method the viscosity with high accuracy is determined through the detection of the driving electric current which resonates the two sensor plates at 30 Hz constant frequency and less than 1 mm of amplitude. The viscosity of the samples was determined at non-physiological (25 ± 1 °C) and ocular physiological temperatures (35 ± 1 °C) [50]. All the measurements were performed in triplicate.

### 2.7. Solubility of Tedizolid Phosphate (TZP)

Solubility of TZP was evaluated by preparing saturated solutions in STF and STF with SLS (0.5%, *w*/*v*). The STF was obtained by dissolving 6.8 g of NaCl, 2.2 g of NaHCO_3_, 1.4 g of KCl and 0.08 g of CaCl_2_·2H_2_O in 1000 mL of purified water [8,51] to get the STF. The excess amount of pure TZP and NC_1_, were put into 1 mL of each medium in triplicate in glass vials. Each mixture was vortexed and put for 72 h in an orbital shaking (100 strokes per min) water bath maintained at 37 ± 1 °C (Julabo^®^ SW22, Seelbach, Germany). After 72 h, the orbital shaking was stopped and the vials were left for 24 h, then centrifuged at 6000 rpm for 20 min to settle down the undissolved solid remains [52]. Thereafter, supernatants were collected, filtered through 0.45 µ filtration unit (Millipore, Molsheim, France) and used to quantify the TZP contents by HPLC-UV method [46] as mentioned in the Appendix A.

### 2.8. In Vitro Release Study

The bioavailability of drugs from any formulation finally depends on the in vitro release of the drug, therefore the release pattern of TZP-NCs was determined. Considering the drug-content the weighed amounts of freeze-dried NC_1_ were suspended in 1 mL of dextrose solution (5%, *w*/*v*) to get 0.1%, *w*/*v* of drug concentration. To evaluate the comparative release profiles, the conventional suspension of TZP (TZP-AqS) was prepared in dextrose solution (5%, *w*/*v*) where polyvinyl alcohol (0.5%, *w*/*v*) and mannitol (1%, *w*/*v*) were added as wetting/suspending and iso-osmotic agents, respectively. The formulation (1 mL of each) was put into the Spectra/Por^®^ Dialysis Membrane (Standard RC Tubing, MWCO 12 KDa). Two ends of dialysis tubing were closed using Spectra/Por^®^ Closures. Formulations containing dialysis tubing were put into beakers (three for each formulation) containing 50 mL release medium (simulated tear fluid (STF, pH 7.4) with 0.5%, *w*/*v*, Sodium Lauryl Sulfate (SLS)). The set of beakers were placed in shaking (100 rpm) water-bath maintained at 35 ± 1 °C (ocular physiological temperature). The dissolution and diffusion of TZP from the tubing in the release medium was assayed by taking 1 mL samples at predetermined time points. Same volume of fresh release medium (kept at 35 ± 1 °C) was added back into the beakers after each sampling to maintain the sink condition and constant volume of the medium. The samples were centrifuged for 5 min at 13,500 rpm. The supernatants were collected and analyzed by HPLC-UV [46]. Drug release (%DR) was calculated by the following Equation (2), and the cumulative amount of TZP released (%) from the formulations was plotted against time. Each formulation was analyzed in triplicate (*n* = 3).
(2)$$\%\mathrm{DR}=\frac{Conc.\;\left(µg/\mathrm{mL}\right)\times Dilution\;Factor\times Volume\;of\;STF\;\left(\mathrm{mL}\right)}{Initial\;dose\;of\;TZP\;used\;for\;the\;experiment\;\left(\mu g\right)}\times 100$$

A model independent mathematical approach was used to compare the release profiles of TZP from NC_1_ and TZP-AqS, the percentage dissolution efficiency (%DE) was used in this study. The calculations of %DEs were performed for each formulation. The mean %DE for the products with 95% CI was compared by measuring the differences between the mean %DE and CI of the two formulations. If the differences of average %DEs and 95% CI are within the set limits (±10%) then the dissolution profiles of TZP-AqS and test (TZP-NC_1_) are considered as equivalent [53,54]. The “%DE’’ was calculated by Equation (3). The calculated %DEs were further analyzed statistically by GraphPad Prism V-5 (GraphPad Software, Inc., San Diego, CA, USA)
(3)$$\%\mathrm{DE}=\frac{\int_{t1}^{t2}Q.dt}{Q_{100\;}\times \;\left(t_{2}-t_{1}\right)}\times 100$$
where, *Q* is the percentage of drug dissolved. DE is the area under the dissolution curve between the time points *t*_1_ and *t*_2_, percentage of maximum dissolution *Q*_100_. The area under the curve is calculated by a model independent method (trapezoidal) by following Equation (4). Where *t_i_* was the *i*th time point, *Q_i_* was the percentage of the dissolved drug at time *t_i_.* Furthermore, the calculated
(4)$$AUC=\overset{i=n}{\underset{i=0}{\sum }}\;\;\frac{\left[t_{1}-t_{ti-1}\right]\;\left[Q_{i-1}+Q_{i}\right]}{2}\;$$

For the determination of release kinetics and mechanism of drug release, the in vitro release data were fitted into different kinetic models such as the zero-order, first-order, Korsmeyer-Peppas and Hixson-Crowell models. From the values of slopes and co-efficient of correlations (*R*^2^) of the kinetic plots obtained by different models, the release-exponent (*n*-value) was calculated, which provided the idea about the drug release mechanism.

### 2.9. Stability Study

The stability study of the NC_1_ was performed following published reports about the nanocrystals with an intention to evaluate the stability of NC_1_ in terms of average size, PDI, ZP, and drug content [16,32,55]. Freeze-dried NC_1_ (10 mg) was packed in different tightly closed amber colored glass containers and stored at 4 ± 2 °C, 25 ± 1 °C and 37 ± 1 °C for 180 days. Alteration in the above parameters were evaluated at 7 days, 30 days, 90 days and 180 days for stability testing of the developed nanocrystals. The stored freeze-dried sample (NC_1_) was redispersed in 5%, *w*/*v* dextrose solution to evaluate the said parameters.

### 2.10. Statistical Analysis

The results of the experiment were presented as mean with standard deviation (±SD), unless otherwise indicated. Statistical analysis was performed using GraphPad Prism V-5 (GraphPad Software, Inc., San Diego, CA, USA). The data were compared by Student’s *t*-test and statistical significance among them was considered at *p* < 0.05.

## 3. Results and Discussion

### 3.1. Formulation Development and Characterization

The process optimization of nano-crystallization was performed by preparing the seven types of TZP-NCs with different stabilizers (Table 1a). Physical characterization (particle-size, polydispersity, and zeta-potential) of the prepared NCs were performed. Considering the required physical characteristics, the batches prepared with PVA and P188 were found better as shown in Table 1a. The better performance of these two stabilizers (P188 and PVA) might be attributed to their higher HLB (>24) values that could avoid the steric barrier between the phases.

Therefore, two batches, consisting of P188 (NC-POL1 to NC-POL4) and PVA (NC-PVA1 to NC-PVA4), were further processed and the results are presented in Table 1b. Among these formulations, NC-POL2 (1% *w*/*v*, P188) and NC-PVA4 (2.5% *w*/*v*, PVA) were found better according to the physical characteristics (Table 1b). The physical characteristics suggested that NCs prepared with 1% *w*/*v*, of P188 had better characteristics than those prepared with 2.5% *w*/*v*, of PVA. This might be due to the good stabilizing property of P188, which could be able to protect the NCs from any steric hindrance. Therefore, NCs prepared using P188 (1% *w*/*v*) were further processed, by changing the durations of homogenization (at 21,500 rpm) and probe sonication (at 40% power) for the optimization purpose. The results of physical characterization and drug content analysis suggested that the NCs-3 batch, which was homogenized for 10 min at 21,500 rpm and sonicated for 60 s (6 cycles, each cycle of 10 s) was the best one (Table 2). The NCs prepared by applying 60 s probe sonication increase their stability by minimizing both the metastable zone and the super-saturation level that is a vital reason for nucleation and crystal growth [47]. Therefore, NCs-3 batch (Table 2) was chosen and freeze-dried with different concentrations of mannitol (as protectant) for further characterization (Table 3). Based on physical characterizations and drug content of the different preparations of NCs-3 batch, the NC_1_ (with 1%, *w*/*v* mannitol) was selected as the best-optimized nanocrystals for the intended purpose. The freeze-dried NC_1_ was further suspended in 5%, *w*/*v* dextrose solution for further physicochemical characterization associated with ocular application.

As we noticed, the prepared NCs (Table 2) had shown the negative zeta-potentials which may obstruct its ocular use. Due to repulsive forces between the negative surface charges on the NCs and negatively charged mucin layer on ocular surfaces, the formulation would not retain for a long time. Therefore, stearylamine (0.2%, *w*/*v* of total formulation) was added to induce positive charge on the surface of NCs [11]. The obtained formulations have shown high magnitudes of positive zeta potentials ranging from +26.3 to +31.6 mV (Table 3). Such positively charged formulations can interact electrostatically with the ocular mucin layer/mucosa that would prolong the ocular retention of the NCs that facilitate its increased transcorneal permeation and ultimately improve the ocular bioavailability of the active form of the drug (tedizolid, TDZ). Also, the high magnitude of positive zeta-potential would provide potential stability to the colloidal system of the NCs.

In this final formulation BKC (0.01%, *w*/*v*) was added as a preservative to avoid any growth of microorganisms introduced unintentionally in the treatment interval. Among different preservatives for ophthalmic preparations, BKC is the frequently used one, especially in multi-dose containers, due to its broad-spectrum antimicrobial property [56]. Apart from preservative action, BKC also stabilizes such ocular NCs. Additionally, being a cationic molecule, BKC resulted in positively charged NCs [7]. Moreover, BKC at 0.1%, *w*/*v* concentration was found virucidal for different adenovirus types [57]. Recently, the broad-spectrum antiviral activity of BKC (in nanodroplet form) against SARS-CoV-2 and other enveloped viruses was also established [58]. Moreover, BKC belongs to a class of quaternary ammonium compounds, therefore, it induced a positive charge to the NCs which was required to improve the electrostatic repulsion among the dispersed NCs (provided stabilization) and to enhance the electrostatic interaction of the NCs with the negatively charged mucin layer of corneal/ocular surface [7]. This interaction would prolong the precorneal/corneal retention of the NCs to provide a sustained release and transcorneal permeation of the drug. These were the reasons why we chose BKC as a preservative in the present investigation.

The antisolvent precipitation is an example of the Bottom-up technique to prepare the NCs, following the principle of controlled precipitation due to the addition of solvent (organic solution containing drug) antisolvent (aqueous solutions of the stabilizers), and evaporation of the organic solvent. The organic solvents were then evaporated by continuous magnetic stirring (12 h) at room temperature to get the suspension of NCs. The supercritical fluid process (SCF) has an advantage over this process in terms of rapid removal of fluids and solvents without requiring extensive drying steps as compared to other solvent precipitation methods. It works with supercritical carbon dioxide (SCO_2_, at temperature ≈31.1 °C and pressure 73.8 bar) for most pharmaceuticals. Due to the low polarity of SCO_2_, the solubilization of lipophilic drugs are easy to form the solution and passing of the drug solution through the capillary tube into an ambient surroundings forms the fine/small particles [9,16]. The SCF technology was successful to improve the flurbiprofen loading into soft contact lens (SCL) and consequently its prolonged release from the lens [59]. Also, the SCF assisted liposomes were prepared for the ocular delivery of ampicillin and ofloxacin [17].

The simple and low-cost instrumentation, low energy consumption, and very low generation of heat are a few important features of the antisolvent precipitation technique. Due to low heat generation, this method was used for heat-sensitive/thermolabile drugs also [47]. Through this method, by applying magnetic stirring, homogenization and sonication could produce the particles/crystals in the submicron to nano-range. Here, precipitation occurs in the region of high turbulence and forceful mixing of the two phases during homogenization and sonication. When the two liquid phases mix with each other, the antisolvent causes the precipitation of TZP as fine crystalline structures, which was also reported previously [60]. In this technique, the selection of stabilizer, speed and duration of solvent-antisolvent mixing, ratio of solvent-antisolvent and of course the ratio of powdered drug/stabilizer and any interaction between these, are important process parameters [61]. The stabilizers with high HLB-values are commonly employed for the preparation of NCs of highly lipophilic drugs [62]. Therefore, we tried PVA, PVP-K30, P188, P407, Vit-E-TPGS, SLS and Tween-80 to prepare the NCs of TZP through this technique. Out of these, P188 and PVA were found to produce well-stabilized and comparatively smaller-sized NCs of TZP as evidenced by the size measurement (Table 1b). Out of the two chosen stabilizers (PVA and P188), P188 was found better one as it produced smaller-sized nanocrystals as compared to the same concentrations of PVA, where we found little larger, elongated and rod-shaped NCs as shown in Figure 2a, which might be due to the inherent property of PVA and comparatively its lower HLB value than that of P188. The unusual, larger, and rosette-shaped structures can be seen in the SEM image of pure PVA at low (7.27 KX) magnification (Figure 2b). The results of SEM imaging were in agreement with the average sizes and polydispersity of the NCs as determined by Zetasizer suggesting that the process and formulation variables for the NCs were optimized in the proper way. The SEM images of pure TZP and TZP-NCs prepared at varying duration of homogenization and probe sonication, respectively are also represented in Figure 3. These images endorsed that the increasing homogenization and sonication time could stabilize the crystals due to relatively smaller size and smooth surfaces. It also justifies that the P188 (1%, *w*/*v*) has better stabilized the NCs in their nano-sized form without changing the crystallinity of TZP, which was also discussed in previous reports [63,64].

Solidification/freeze-drying was performed to provide stabilization to the NCs, as we know that solid formulations are more stable than their liquid forms. The freeze-drying of suspensions of NCs would reduce the unstable factors (Ostwald ripening and aggregation) of NCs. Thus, the suspensions of NCs are converted into dried forms and the dried crystals are converted into different dosage forms including sterile powder for injection/ophthalmic, capsules and oral tablets [65]. The aggregation/growth of NCs should be curtailed during freeze-drying. In the suspension of NCs, the use of stabilizers offers spatial or ionic stability to the drug NCs by becoming adsorbed onto their surfaces and preventing the occurrence of any unstable phenomenon (Ostwald ripening or aggregation), thereafter the dried products must exhibit acceptable dispersion ability in case of contact with aqueous phase [66,67]. But, the process of drying may result in the solidification of the stabilizer also, which may cause an irreversible aggregation of the NCs [66]. Therefore, the use of protectants during freeze-drying has become necessary to prevent the occurrence of such unusual phenomenon. Thus, mannitol (1%, *w*/*v*) was added as a protectant to the suspension of NC_1_ before freeze-drying to get satisfactory re-dispersible freeze-dried NCs [68].

Overall, the results of the present study suggested the antisolvent precipitation method was suitable for the nano-sizing of TZP (a poorly soluble drug) in the range of 150.4–163.2 nm (Table 3). The present method can be applied as an alternative to flash precipitation by CLIJ for the other hydrophobic and poorly soluble drugs such as ibuprofen, salbutamol sulphate, amphotericin-B and cyclosporine-A to get the NCs of sub-micron sizes [47,69,70]. As compared to CLIJ where mixing occurs only once, here if nucleation and crystal growths remain incomplete, secondary crystallization may occur for further growth of the NCs. The antisolvent-precipitation (Bottom-up technique) can further be advanced by employing other “Generally Regarded as Safe” excipients (ophthalmic preparations) for the successful completion and execution of TZP-NCs for ocular use.

### 3.2. Size, Polydispersity-Index (PDI) and Zeta Potential (ZP)

The physical characterization including the size, PDI and ZP of the preliminary batches of TZP-NCs prepared with varying stabilizers (Table 1a) and their optimization using Poloxamer-188 and PVA (stabilizers) are summarized in Table 1b. Out of different stabilizers the NCs prepared with P188 and PVA have shown relatively lower crystal sizes 495.3 ± 23.5 nm (with PDI and ZP values of 0.265 ± 0.011 and −16.23 ± 2.8 mV) and 514.8 ± 29.4 nm (with PDI and ZP values of 0.375 ± 0.023 and −6.8 ± 1.9 mV), respectively (Table 1a). The optimization of the TZP-NCs with P188 and PVA indicated that the NCs coded as NC-POL2 (with 200 mg of P188) have shown (Table 1b) comparatively smaller size (481.7 ± 58.4 nm) with a low value of PDI (0.252 ± 0.021) and slightly high negative ZP (−17.2 ± 5.6 mV).

Further optimization of the NCs prepared with Poloxamer-188 (1%, *w*/*v*) at varying durations of homogenization (21,500 rpm) and probe sonication without mannitol further reduced the size of the NCs (Table 2). The characterization of the four optimal NCs (NCs−1 to NCs−4), suggested that the smallest NCs (150.4 ± 18.3 nm) were obtained by homogenization for 10 min at 21,500 rpm followed by probe sonication for 60 s (NCs−3 in Table 2). The PDI of NCs−3 was lower (0.231 ± 0.006) and zeta potential was relatively higher (−13.5 ± 1.5 mV) with the highest drug content (96.7 ± 1.2%).

### 3.3. Effect of Homogenization and Probe Sonication Time Duration on Size and PDI

The homogenization and probe sonication time duration had a determinant effect on the crystal size. Initially, the crystal size was 565 nm prepared by homogenization (5 min) and probe sonication for 40 s with a PDI of 0.428. Increasing the homogenization and probe sonication to 7.5 min and 50 s, respectively, the size was reduced to 389 nm with PDI of 0.401 and at 10 min and 60 s homogenization and sonication, the smaller sized crystals (150 nm) were obtained with the lowest value of PDI (0.231). Further increase in time duration for homogenization and sonication to 15 min and 70 s, respectively, could not cause a significant reduction in the crystal sizes (147 nm), although a slight but not significant increase in PDI (0.239) was observed. Thus, the homogenization for 10 min followed by probe sonication for 60 s, was sufficient to get the required sized NCs of TZP without any agglomeration and crystal growth.

### 3.4. Effect of Mannitol Concentrations on Size, ZP and PDI

The hydrophobic interaction during the freeze-drying may agglomerate the NCs. Hence, for proper re-dispersion of the NCs after freeze-drying, the addition of the cryoprotectant was needed. Here, we have used different concentrations of mannitol (1–5%, *w*/*v*) as shown (Table 3). Although no significant changes in the size, ZP and PDI of NC_1_ were observed at all concentrations of mannitol, but the highest drug content remained almost constant (96%) with slightly increased negative ZP at 1%, *w*/*v* mannitol. The slight increase in size of NC_1_ at most of the concentrations was due to the aggregation of the crystals. Such aggregation was attributed to occurrence of the capillary forces (as per the capillary pressure theory) during the process of freeze-drying. Thus, mannitol (1%, *w*/*v*) was optimal to prevent the aggregation/crystal growth and provided stabilization to NC_1_ during freeze-drying.

Thus, the NCs-3 batch was freeze-dried with different concentrations of mannitol (Table 3). Out of four formulations listed in Table 3, the NC_1_ (freeze-dried with mannitol) was considered as the best optimized nanocrystals for ocular use. The smallest size (154.3 nm), low PDI (0.243) and highest positive (due to stearylamine and BKC) ZP (+31.6 mV) of NC_1_ even after freeze-drying encouraged us to choose the NC_1_ for further studies including the physicochemical characterizations for ophthalmic products. The low PDI indicated the unimodal distribution and its highest ZP could provide the best colloidal stability to the suspended NCs (NC_1_). The size and zeta potential distributions of the different NCs and the optimal formulation (NC_1_) were represented in Figure 4.

### 3.5. DSC Analysis

The DSC curves obtained for pure TZP and other components with their PM and NC_1_ were represented in Figure 5. The curve of TZP (Figure 5a) exhibited a sharp endothermic peak at 206.3 °C, which corresponds to its melting point and is considered typical for any crystalline and anhydrous molecule [71]. The curve of P188 (Figure 5b) has shown a very broad endothermic peak at 66 °C, which was around its reported melting point [64,72]. In the curve of mannitol (Figure 5c), a slightly broad endothermic peak was seen at 177.5 °C which was near its reported melting point [73]. In the curve of physical mixture of TZP, P188 and mannitol (Figure 5d), three separate and almost unchanged endothermic peaks were located near their respective melting points as described above which were indicating the crystalline nature of the individual component in the investigation.

In the curve of NC_1_ (Figure 5e), the endothermic peak of TZP appeared slightly at a lower temperature (199.6 °C) compared to the melting temperature of pure TZP (206.33 °C), indicating the crystalline character of TZP was not altered during its nano-crystallization by homogenization followed by sonication. The reduced melting temperature of TZP in NC_1_ is attributed to its nano-crystallization (reduced crystal-size) and decrease in the crystal lattice energy of TZP. The decrease in the enthalpies was the consequence of the interaction and incorporation of TZP molecules in the hydrophobic territory of micelles created by high HLB (HLB-29) of P188 [74], which was also reported during the nano-crystallization of atorvastatin using Poloxamer-188 as stabilizer [64]. Moreover, a less intense endothermic peak was appeared (near to 177.5 °C) in the DSC curve of NC_1_, which was very near to the melting temperature of mannitol, indicating the presence of mannitol in the nanocrystals (NC_1_) and no endothermic peak around the melting temperature of P188 was observed, which confirmed the proper washing of the NC_1_ to remove the extra surfactants. The results of DSC validate the decreased crystallinity of TZP in NC_1_ form which potentiates its solubility. The increased solubility would improve the transcorneal permeation and hence would increase the ocular bioavailability of the drug.

### 3.6. FTIR Analysis

The FTIR spectrum of TZP (Figure 6a) has significant vibrations of C-H wagging at 877.8 cm^−^^1^, C=C stretching as well as C-C stretching of Phenyl ring at 1620.1 cm^−^^1^, symmetric C-H stretching in Phenyl ring of (Pyridine-3-yl) Phenyl-3-Fluoro structure at 3256.4 cm^−1^, C-O stretching vibrations at 1209.2 cm^−^^1^, C-H wagging at 1325.5 cm^−^^1^, C-H wagging as well as C-N-C bending vibrations at 1407.3 cm^−1^, C=O stretching vibrations) in the 1,3-oxazolidin-2-one structure at 1746.4 cm^−^^1^ and C-H wagging in Pyridine ring and C-C-N bending vibrations in the (Pyridine-3-yl) Phenyl-3-Fluoro structure at 1147.8 cm^−1^ and O-H (at 5C position of Oxazolidinone ring) stretching at 3256.4 cm^−1^ [71,75].

The spectrum of P188 (Figure 6b) has shown characteristic and principal absorption peaks of C-H stretching of aliphatic structure at 2875.2 cm**^−^**^1^, in-plane O-H bending vibration at 1344.1 cm**^−^**^1^ and C-O stretching at 1099.3 cm**^−^**^1^. An overlapping and unnoticed shifting of distinct C-O stretching of P188 at 1078.2 cm**^−^**^1^ was noted in PM, indicating no change in its functional group during the preparation of PM. The FTIR spectrum of mannitol (Figure 6c), has shown the characteristic absorption bands of C-H stretching at 1413 cm**^−^**^1^ and C-O stretching at 1011.6 cm**^−^**^1^ as well as at 1075.1 cm**^−^**^1^. The distinguished absorption band of mannitol was as observed at 1011.6 cm**^−^**^1^ and 1075.1 cm**^−^**^1^ for C-O stretching which were also noted at 1012.8 cm**^−^**^1^ as well as at 1078.2 cm**^−^**^1^ (slightly shifted to higher frequency) in the PM, indicating no change in its molecular structure. The presence of major absorption peaks of TZP, P188, and mannitol in the PM of the excipients with TZP as shown in Figure 6d, indicates that the crystalline behavior of the molecules was not changed in the preparation of PM, which was also reported previously [76]. Figure 6e, has shown the FTIR spectrum of NC_1_. Among the distinguish bands obtained for the pure TZP (as shown in Figure 6a), the spectrum of the NC_1_ did not show any shifting or less intense bands (except at 3256.4 cm**^−^**^1^) at 877.8 cm**^−^**^1^, 1105.4 cm**^−^**^1^, 1208.4 cm**^−^**^1^, 1406.8 cm**^−^**^1^, 1464.1 cm**^−^**^1^, 1619.8 cm**^−^**^1^ and 1746.4 cm**^−^**^1^ (characteristic of the molecular structure of 1,3-Oxazolidinone) and (Pyridine-3-yl) Phenyl-3-Fluoro of TZP (Figure 5e). These characteristic bands were predominantly associated with the stretching vibrations of the C-C, C-O and C-F sigma bonds as well as the C-H wagging [75].

However, there was a shifting of O-H (at 5C position of Oxazolidinone ring) stretching (at 3256.4 cm^−1^) as present in the spectrum of TZP. It was shifted towards a lower frequency (2880.2 cm^−1^) in the spectrum of NC_1_. The reason for this might be due to the formation of hydrogen bonding between hydrogen and oxygen molecules of TZP, P188 and mannitol, which was also reported during the complexation of TZP with *β*-cyclodextrin [71] and preparation of solid dispersion of aceclofenac with mannitol and hydroxyl *β*-cyclodextrin.

The characteristic in-plane O-H bending vibration at 1344.1 cm^−1^ and C-O stretching at 1099.3 cm^−1^ for P188 were not present in the final formulation (NC_1_), which was due to the proper washing of the NC_1_ by Milli-Q water by centrifugation to remove the extra surfactants. Finally, the results of FTIR spectroscopy confirm that the nano-crystallization of TZP, did not alter the basic molecular structure of TZP during the preparation of NC_1_ in presence of a high HLB-value (HLB-29) of P188.

### 3.7. XRD Analysis

The overlay X-ray diffractograms of TZP, P188, Mannitol, Physical mixture of drug, P188 and Mannitol (PM), and the optimized formulation (NC_1_) were represented in Figure 7. The diffractogram of unprocessed and pure TZP (Figure 7a) has shown some characteristic sharp diffraction peaks at 2*θ* values of 14.4°, 23.8°, 38.1° and 44.3° having intensities of 3490 cps (with Bragg’s or *d*-value 6.145 and I/I_0_ 100), 2526 cps (with *d*-value 3.735 and I/I_0_ 73), 2492 (with *d*-value 2.4 and I/I_0_ 72) and 1036 (with *d*-value 2.04 and I/I_0_ 30) clearly indicating the crystallinity of the drug. The diffractogram of P188 (Figure 7b) has characteristic intense diffraction peaks at 2*θ* values 19.4° and 23.5° with intensities of 3434 cps (with *d*-value 4.6 and I/I_0_ 96) and 3583 cps (with *d*-value 3.8 and I/I_0_ 100) suggesting its crystallinity. In Figure 7c, the diffractogram of mannitol is representing the peaks at 2*θ* values 15.0°, 19.1°, 21.4°, 23.8°, 29.8° and 39.0° with intensities 1814 cps (with *d*-value 5.9 and I/I_0_ 36), 4060 cps (with *d*-value 4.6 and I/I_0_ 80), 1861 cps (with *d*-value 4.1 and I/I_0_ 37), 5092 cps (with *d*-value 3.7 and I/I_0_ 100), 1177 cps (with *d*-value 2.9 and I/I_0_ 24) and 1037 cps (with *d*-value 2.308 and I/I_0_), suggesting the crystalline nature of mannitol. The diffractogram of PM (Figure 7d) has shown sharp diffraction peaks at 2*θ* of 14.4°, 19.3°, 23.8° and 26.4° with intensities 1586 cps (with *d*-value 6.1 and I/I_0_ 46), 3472 cps (with *d*-value 4.6 and I/I_0_ 100), 2978 cps (with *d*-value 3.8 and I/I_0_ 86) and 512 cps (with *d*-value 3.7 and I/I_0_ 15), indicating the crystallinity of the individual constituent in their physical mixture.

The diffractogram of NC_1_ (Figure 7e) has sharp diffraction peaks at approximately the same 2*θ* values for the pure drug (TZP) with slightly lower intensities 1348 cps (with *d*-value 9.1 and I/I_0_ 89), 1531 cps (with *d*-value 4.3 and I/I_0_ 100), 626 cps (with *d*-value 4.1 and I/I_0_ 41), 608 cps (with *d*-value 3.6 and I/I_0_ 40), 392 cps (with *d*-value 2.4 and I/I_0_ 26) and 369 cps (with *d*-value 2.2 and I/I_0_ 25).

The presence of characteristic diffraction peaks of TZP in the diffractogram of NC_1_ suggests that the formulation factors (homogenization followed by sonication) could not alter the crystallinity of TZP, which were also observed in previous studies [64,76,77,78]. However, reduction in the peak intensities were noted around the same 2*θ* values suggesting the reduced crystalline characters of TZP in NC_1_ form, which might be associated with the surface coverage and modification of TZP in presence of P188. Also, some extra sharp diffraction peaks were found with the diffractogram of NC_1_, those were due to the presence of P188 and the use of mannitol (cryoprotectant) during freeze-drying of the NC_1_.

The crystalline sizes of TZP-NC_1_ and TZP-pure were calculated by the Scherrer equation as mentioned above using the software Origin© V-7.0 (OriginLab Corp., Northampton, MA, USA). Considering the morphology of TZP-crystals elongated rod shaped, the Scherrer constant value was 0.68 used for the calculation. The crystal size of TZP-NC_1_ was 113.4 ± 25.2 nm. Which were almost similar to the size measurement by zeta-sizer as mentioned above. The results of XRD studies confirm the decreased crystallinity of TZP in NC_1_ form which was substantiated by the increased saturation solubility of the NC_1_, in simulated tear fluid with SLS.

### 3.8. Physicochemical Characterization

The results of physicochemical characterization for NC_1_ are presented in Table 4. The visual examination of NC_1_ suspension under normal light against a dark and white background was found clear and transparent. So, the formulation will not cause any blurring of vision and will be appropriate for ocular application. The pH of human tear fluids ranged from 6.5 to 7.6 with a mean value of 7.0 as measured by Abelson [79]. In the present investigation the pH of NC_1_ was found 7.0 ± 0.4 which falls within the normal range. The rate of tear turnover and the chemical buffering action/capacity of tear fluids can easily bring the pH of the formulation (NC_1_) to its own pH by the neutralization process of tear fluids. So, the formulation will not cause any discomfort to the eyes even at unstable tear-film due to bacterial lysis of long-chain components of Meibomian lipids (as stable composition of cholesterol and cholesteryl esters) to free fatty acids that cause irritation to the tear film and the corneal surfaces [80].

The osmolarity of the NC_1_ suspension was found around 298 ± 5 mOsm·L^−1^, which was almost equivalent to the osmolarity of normal tear fluid. The normal tear fluid has an osmolarity of 300–302 mOsm·L^−1^ (iso-osmotic) [80], but it may increase up to 320–340 mOsm·L^−1^ (hyperosmolarity) in some ocular conditions. Such as bacterial infections or in dry eye conditions, the tear film has shown an increased osmolarity (hyperosmolarity) [81]. Therefore, the topically applied ophthalmic preparations for such diseased conditions should be formulated as hypotonic to counteract the increased osmolarity of the tears/tear film [82]. The conversion of freeze-dried samples of NC_1_ to aqueous suspension using dextrose solution (5%, *w*/*v*) as a vehicle did not alter the drug content of the optimized nanocrystals significantly (*p* < 0.05). This indicated a temporary stability of the drug in its aqueous suspension form. The viscosity of NC_1_ suspension at 20 ± 1 °C was 28.5 ± 2.4 cPs and at ocular physiological temperature (35 ± 1 °C), it was 21.1 ± 1.1 cPs. The viscosity of the optimized formulation falls within the limit of desired viscosity for ophthalmic preparations (25–50 cPs). A slight decrease in viscosity at 35 ± 1 °C (attributed to high shear stress at increased temperature), suggesting that the suspension of NC_1_ could easily be distributed throughout the ocular surface without any irritation/discomfort in or around the eyes of the patient even during blinking of eyes [83]. Thus, NC_1_ would be convenient and stress-free for its ocular application.

### 3.9. Solubility Determination

The saturation solubility of pure TZP was 10.8 ± 2.4 µgmL^−1^ and 16.1 ± 3.8 µgmL^−1^ in STF and STF with 0.5%, *w*/*v* of SLS, respectively, while a notable increased solubilization of TZP was found from nanocrystals (NC_1_) form, which was 18.4 ± 2.4 µgmL^−1^ and 25.9 ± 3.1 µgmL^−1^ in STF and STF with 0.5%, *w*/*v* of SLS, respectively. The results of this study revealed that particles/crystals in nano-size have shown a significant (*p* < 0.05) increase in the saturation solubility of TZP. The improved solubilization of the drug was attributed to the fact that nano-sizing (using P188) of the particles lead to an increase in the overall surface area to interact with the aqueous phase. Thus, the point of contact for the drug particles with the solvents (STF and 0.5%, *w*/*v* of SLS) were increased, which facilitated the wetting and dispersibility of drug particles [84] and hence increased the solubilization of TZP in NC_1_. This finding correlates with the well-known “Noyes Whitney Equation” that defines the reliance on comparative saturation solubility of different particles with varying radii and concentration of the dissolved solute [85].

Furthermore, the augmented solubilization of TZP was due to a strong affinity between TZP and P188 to produce “molecular dispersion” which is accountable for altering the solubility equilibrium and saturation solubility of TZP. This finding was substantiated by a previous report on the nano-sizing of poorly aqueous soluble atorvastatin with Poloxamer-188 [64]. The improved solubilization of TZP in STF with SLS would facilitate the in vitro release profiling of the drug. Also, an increased solubility of TZP would improve the oral or ocular bioavailability of the active form of TZP (i.e., tedizolid).

### 3.10. In Vitro Release Study

In vitro drug release study confers the practical consideration about the possible in vivo performance of any developed formulation. Therefore, this experiment was carried out in simulated tear fluid (STF) with SLS (0.5%, *w*/*v*) for the release of TZP from the freeze dried NC_1_ and TZP-AqS. The SLS was added in STF to increase the solubility of the poorly soluble drug (TZP). The release profile of the two formulations is represented in Figure 8. The assessment of the obtained release profiles clearly shows a significant (*p* < 0.05) enhancement in the release rate of NC_1_. The TZP-nanocrystals (NC_1_) have shown a fast release of ~15.3% at 1 h and ~49.2% drug was released within 4 h, followed by delayed release till 12 h (~78.8%). While the pure TZP from AqS has shown only ~6.3% at 1 h and ~24.2% drug was released within 4 h and ~43.1% till 12 h. Initially, the increased rate of drug release from the nanocrystals was contributed by the smaller sized (151.6 to 154.3 nm) NC_1_ that provided an overall increased surface area. Due to increased surface area, a large portion of the nanocrystals could come in contact with the release medium, causing higher solubilization and dissolution of the TZP which in turn provided its higher saturation solubility [55].

Moreover, the stable smaller size of TZP-nanocrystals prevented their aggregation and facilitated the surface wetting ability because of the presence of traces of P188 and hence the dispersibility of the drug [64]. The presence of P188 at the interface of TZP and STF with SLS (0.5%, *w*/*v*) also facilitated the reduction in their interfacial tension by the interaction of ether-oxygen (R–O–R′) of hydrophilic PEO-blocks of P188 through H-bonding with water molecules [86]. On the other hand, the linearity in release profile (till 12 h) of TZP from NC_1_ might be endorsed due to the multimolecular micellar formation of P188, which was also reported in the in vitro release of atorvastatin (a poorly aqueous soluble drug) in the nanocrystal states [64]. The central PPO-block of P188 that is the hydrophobic part of the micellar structure might interact with TZP through Van-der Waals intermolecular forces and decrease the partitioning and diffusion of TZP from the core of multimolecular micellar structures [87].

From the dissolution data, the calculation of %DE with their respective 95% CI are illustrated in Figure 9. The average %DE for TZP-NC_1_ and TZP-AqS were found to be 70.6 ± 3.4% and 56.2 ± 3.5% with mean dissolution times 2.9 ± 0.2 h and 4.3 ± 0.3 h, respectively. The 95% CI was found in the range of 13.3 to 15.8 (>±10%), indicating the differences in the dissolution/release profiles of the drug from the two formulations. Calculation of %DE provides information for the quantitative comparisons between the products. It is easier to interpret the data obtained through %DE than that of the difference and similarity factors [53]. Conclusively, the %DE is a better substitute for the single point dissolution measurement for the dissolved drugs during in vitro release study.

Application of some kinetic models on the in vitro release data we found that the in vitro release of TZP from both the formulations followed the first-order release kinetics. Although the release kinetics of the drug from the formulations was the same, but the release of TZP was linear till 12 h from the NC_1_ while it was linear up to 8 h from the TZP-AqS, thereafter the drug release remained almost the same till 12 h. Among the applied kinetic equations or release models the highest correlation coefficients (*R*^2^) were 0.9875 and 0.9645 (for TZP-NC_1_ and AqS of TZP-pure, respectively) (Appendix A), which were associated with the first order model. The release kinetic parameters for the two formulations after applying the kinetic models, were mentioned in Table 5. By taking the *R*^2^ values and slope of the tried kinetic equations, the diffusion-exponents (*n*-value) were obtained. The *n*-values (0.0238 and 0.0081 for TZP-NC_1_ and AqS of TZP-pure, respectively) indicate the Fickian-Diffusion type of release mechanism.

### 3.11. Stability Studies

The optimized nanocrystals (NC_1_) were evaluated at stipulated time points for the physical and chemical stabilities of NC_1_ to ascertain the stability limits in support of its storage at 4 ± 2 °C, 25 ± 1 °C and 37 ± 1 °C for 180 days. The results were summarized in Table 6. The results of physical stability did not show notable alteration in PDI, zeta-potential, TZP content and cumulative amount of drug released. No significant alteration in the selected parameters were observed at 4 ± 2 °C, but a slight increase in size at 90 days and 180 days, was observed in the samples stored at 25 ± 1 °C and 37 ± 1 °C. Similarly the increased size was also reported for the stability of liposomes stored at different temperatures for 150 days [16]. The crystal growth in the aqueous environment is generally attributed due to the “Ostwald ripening”, where the smaller crystals in the aqueous medium might get dissolved and deposited on to the larger crystals to minimize the surface/area ratio for achieving the thermodynamic stability [64,88]. But the size growth of NC_1_ in the present investigation stored in freeze-dried state at slightly higher temperatures was not because of the “Ostwald ripening” phenomenon, rather it might be due to dehydration of the stabilizer (P188) and the protectant (mannitol) and consequent loss of NC_1_ protection [89]. Unremarkable alterations in the PDI, ZP and drug content was noticed even at higher temperature for 180 days. The drug content analysis indicated that the TZP content was around 94.9% up to 180 days, demonstrating that the optimized freeze-dried TZP-nanocrystals (NC_1_) remained stable without any significant degradation of the drug. Moreover, the results obtained for drug content suggested that the homogenization (at 21,000 rpm) followed by probe sonication (at 40% power for 60 s) processes to obtain the NC_1_ did not affect the chemical stability of TZP [90,91]. Overall, the TZP-NC_1_ were physically (size, polydispersity-index and zeta potential) and chemically (drug content) stable at all three storage temperatures (4 °C, 25 °C and 37 °C). Thus, the product can be stored at these storage temperatures without significant alterations in the above-mentioned physical and chemical parameters for up to 180 days.

## 4. Conclusions

The positively charged (+29.4 mV) TZP-NCs prepared by antisolvent precipitation method including the homogenization followed by sonication, resulted in considerably smaller sized (≈151.6 nm) nanocrystals (suitable for ocular use). The NCs were well stabilized by Poloxamer-188 at 1% (*w*/*v*) concentration. Mannitol (as cryoprotectant at 1%, *w*/*v*) prevented the crystal growth and stabilized the TZP-NC_1_ during freeze-drying. The SEM study indicated good crystalline morphology of the optimal NCs. The FTIR spectroscopy revealed that the basic molecular structure of TZP was not altered, while the DSC and XRD studies showed a reduction in the crystallinity of the drug in NC_1_ form. Due to reduced crystallinity and nano-sizing of TZP, the solubility of the drug was increased by 1.4-times as compared to pure-TZP in STF with SLS (0.5%, *w*/*v*). The increased solubility of TZP-NC_1_ would definitely improve the transcorneal permeation which in turn would increase the ocular bioavailabity of the active form of the drug (TDZ) if applied in vivo. The redispersion of freeze-dried NC_1_ in dextrose solution (5%, *w*/*v*) with mannitol (1%, *w*/*v*) ensued in a clear transparent iso-osmotic nano-suspension with ≈298 mOsm·L^−1^ osmolarity and ≈21.1 cps viscosity at 35 °C. Relatively higher drug release (≈78.8%) was found from NC_1_ at 12 h as compared to TZP-AqS (≈43.4%) during in vitro release study. %DE determination indicated a different release profile of the drug from the two formulations. The TZP-NC_1_ were physically (size, PDI and ZP) and chemically (drug content) stable at all three storage temperatures (4 °C, 25 °C and 37 °C) for 180 days. Based on the above findings, the TZP-NC_1_ would be a promising and viable alternative for the ocular delivery of TZP in vivo. Further studies including in vitro antimicrobial study in vivo ocular irritation and pharmacokinetics in rabbits have been performed to ensure the efficacy, safety, and drug bioavailability but these are out of the scope of the present article.

## Figures and Tables

**Figure 1 pharmaceutics-14-01328-f001:**
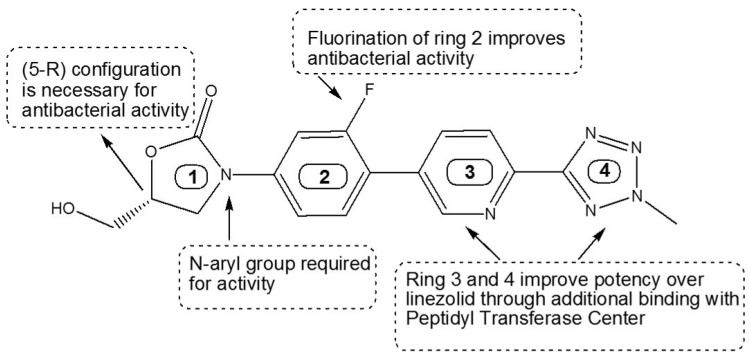
The chemical structure of TDZ, where 1–4 symbolizes the different aromatic rings present in the structure of TDZ. Ring-1 represents the oxazolidinone nucleus; Ring-2 is the meta-fluorine and para-oriented electron withdrawing group; Ring-3 is the pyridine ring and Ring-4 is the tetrazole ring.

**Figure 2 pharmaceutics-14-01328-f002:**
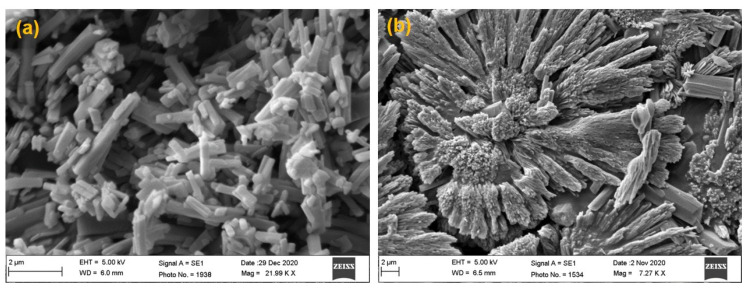
SEM images of elongated and rod-shaped NCs (**a**); unusual, larger, and rosette-shaped NCs (**b**). Both these NCs were stabilized by PVA (1%, *w*/*v*) alone.

**Figure 3 pharmaceutics-14-01328-f003:**
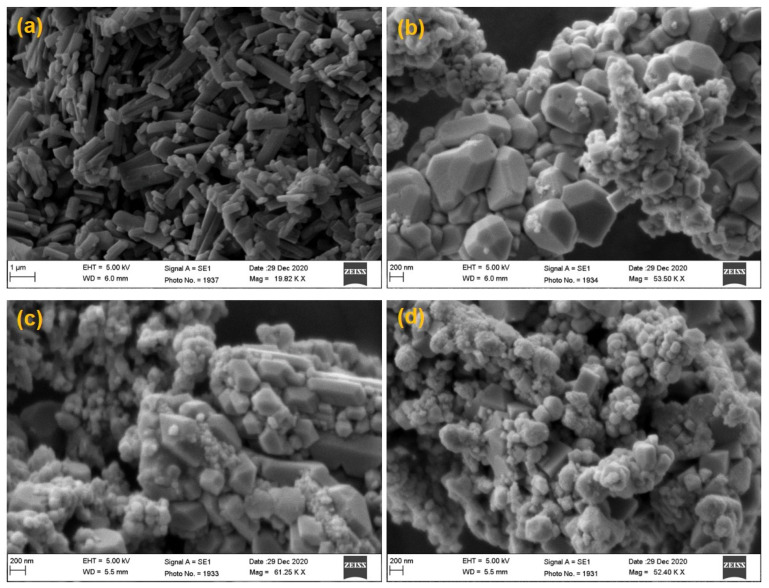
SEM images of pure tedizolid phosphate powder (**a**); and TZP-NCs prepared at varying duration of homogenization and probe sonication, respectively: 5 min and 40 s (**b**); 7.5 min and 50 s (**c**); 10 min and 60 s (**d**).

**Figure 4 pharmaceutics-14-01328-f004:**
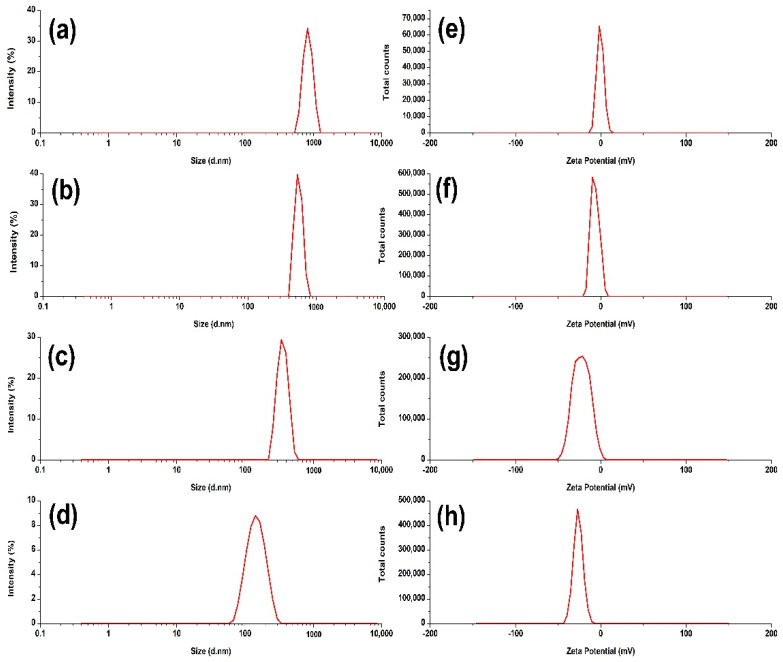
The size distributions of NCs−1, NCs-2, NCs−3 and the optimized NCs−3 as NC_1_ ((**a**–**d**), respectively). Zeta potential distributions of NCs−1, NCs−2, NCs−3 and the optimized NCs−3 as NC_1_ ((**e**–**h**), respectively).

**Figure 5 pharmaceutics-14-01328-f005:**
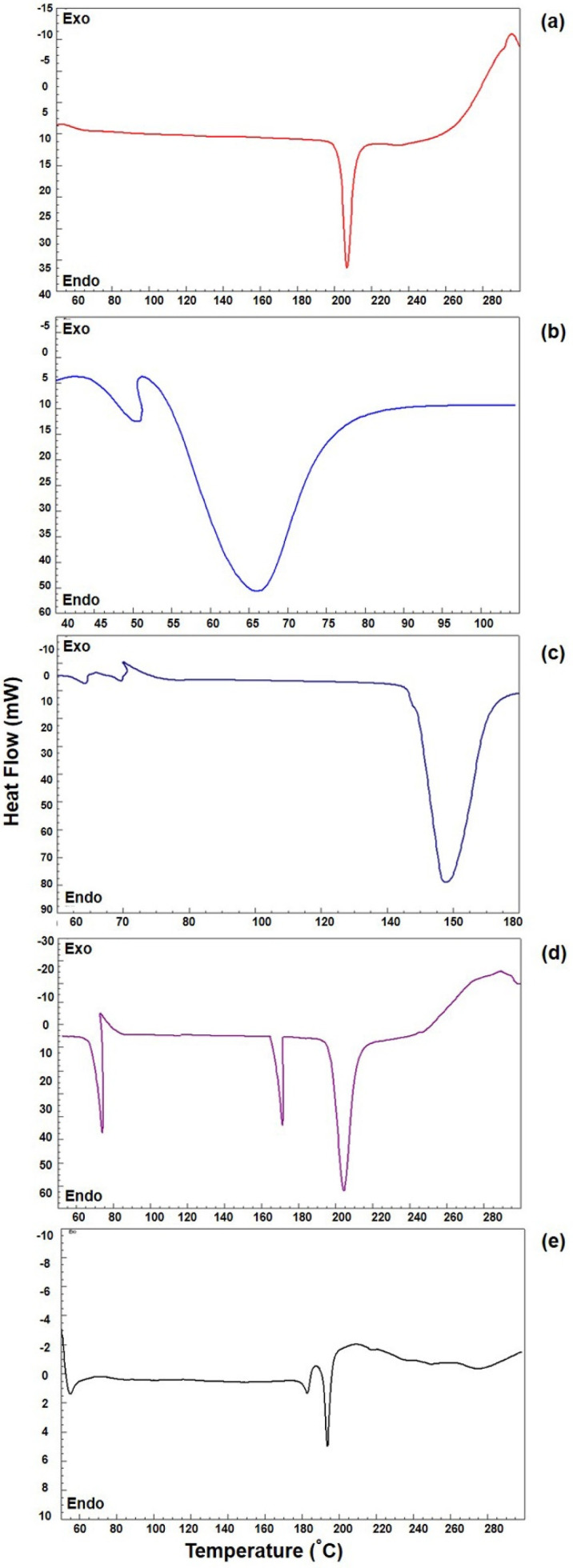
DSC−endotherms of TZP (**a**), P188 (**b**), mannitol (**c**), physical mixture (PM) of TZP, P188, and mannitol (**d**) and lyophilized NC_1_ (**e**).

**Figure 6 pharmaceutics-14-01328-f006:**
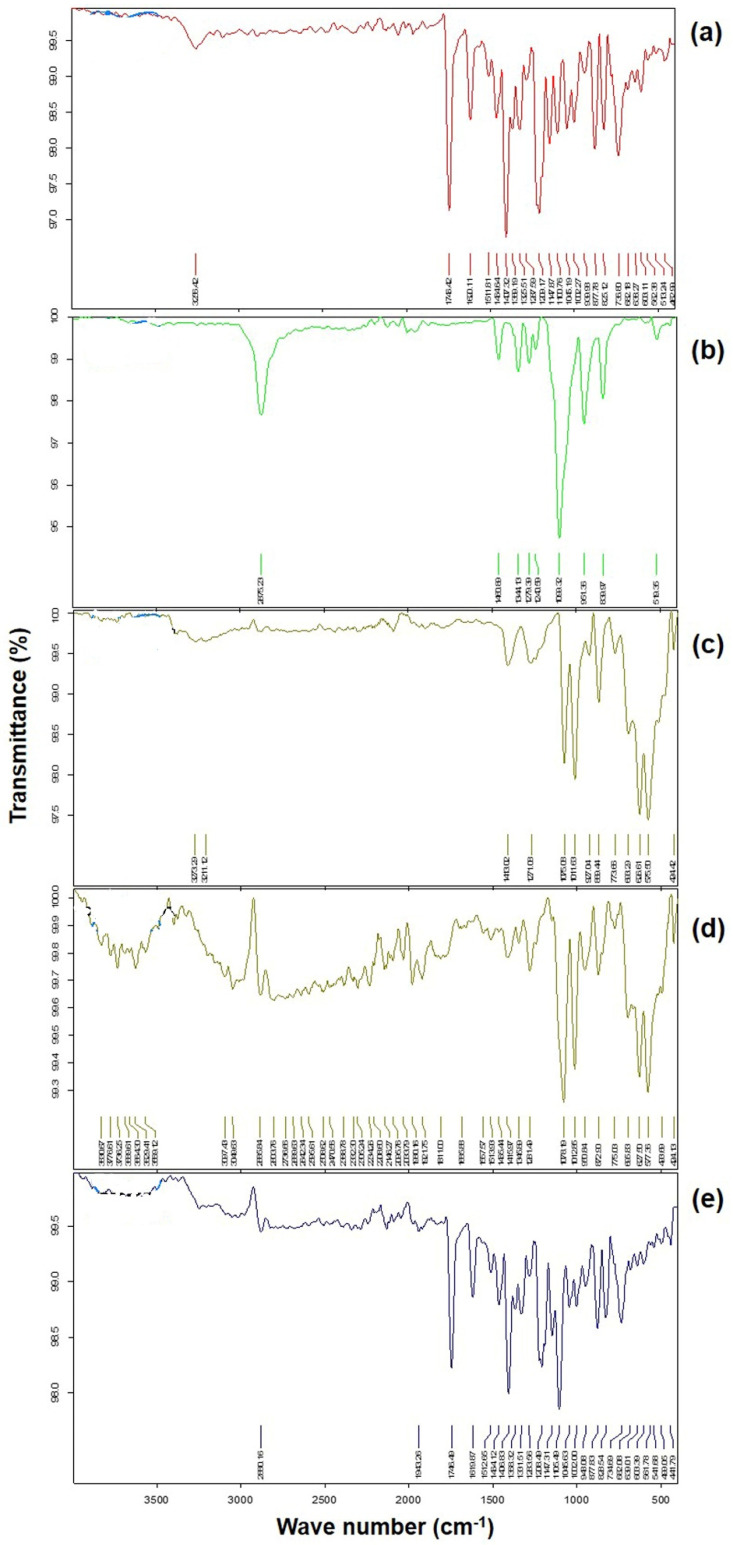
FTIR−Spectra of pure TZP (**a**), P188 (**b**), mannitol (**c**), PM of TZP, P188, and mannitol (**d**) and lyophilized NC_1_ (**e**).

**Figure 7 pharmaceutics-14-01328-f007:**
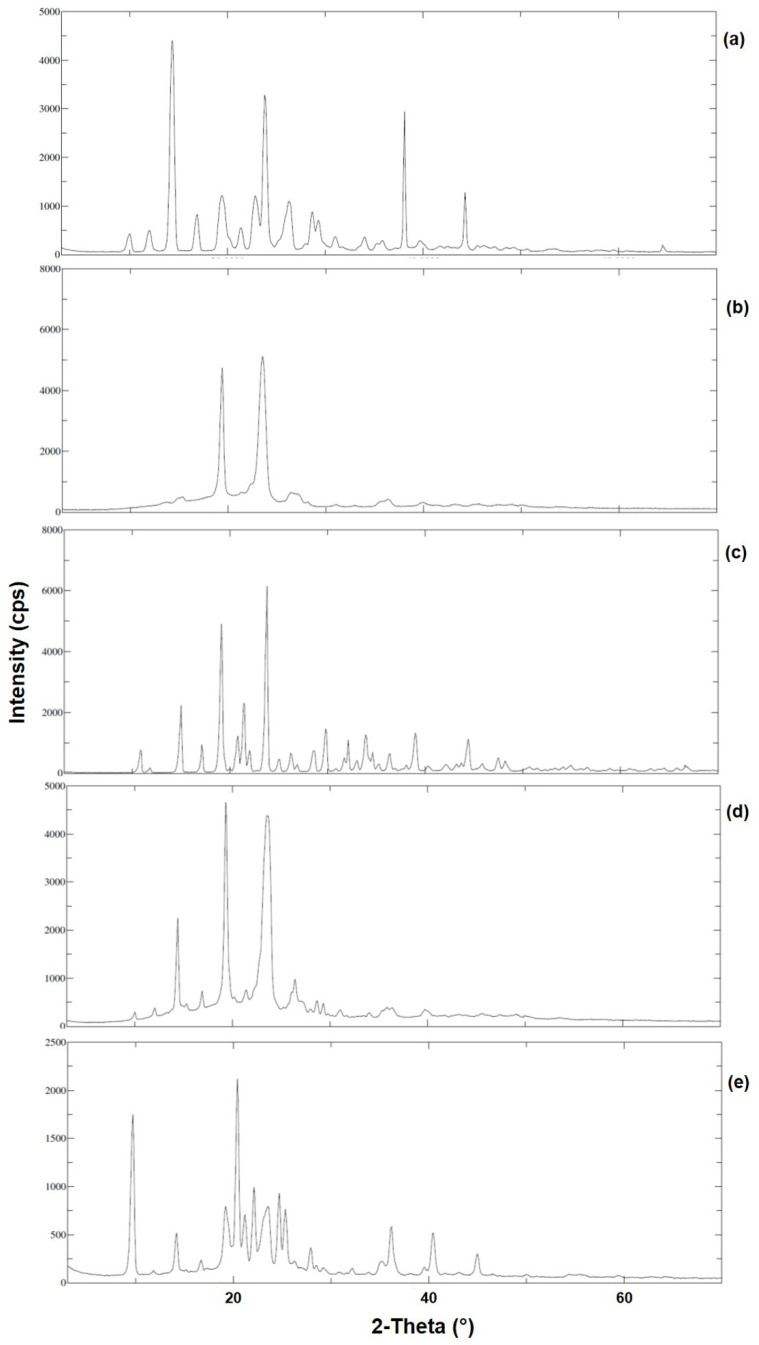
XRD patterns of pure TZP (**a**), P188 (**b**), mannitol (**c**), PM of TZP, P188 and mannitol (**d**) and lyophilized NC_1_ (**e**).

**Figure 8 pharmaceutics-14-01328-f008:**
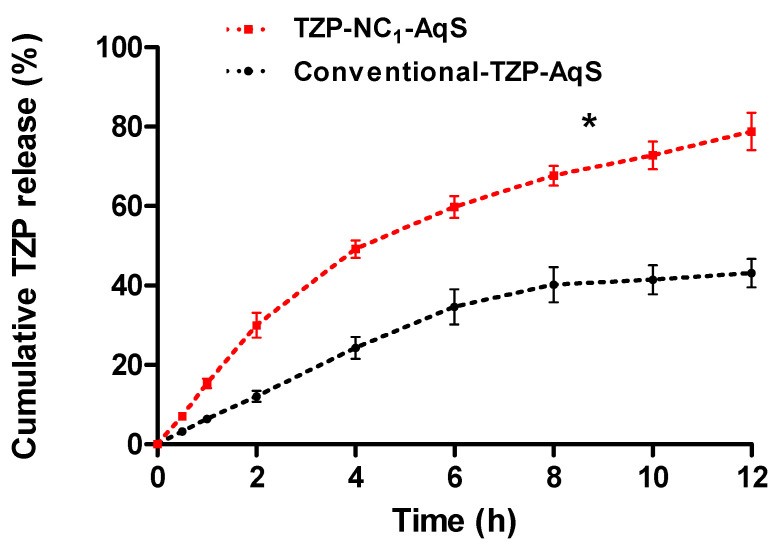
In vitro release of TZP from the freeze dried TZP−NC_1_−AqS and conventional TZP−AqS in STF (pH 7.0) with SLS (0.5%, *w*/*v*). Results are the mean with ± SD of three measurements and “*” *p* < 0.05; TZP−NC_1_ versus TZP−AqS.

**Figure 9 pharmaceutics-14-01328-f009:**
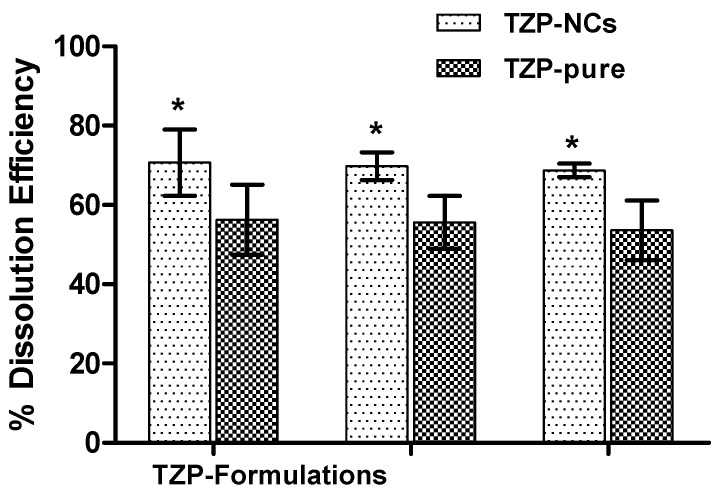
Dissolution efficiencies (%DE) with 95% confidence intervals of TZP-NCs versus TZP−AqS, where * *p* < 0.0005 was indicating the significant difference between the %DE of the two formulations.

**Table 1 pharmaceutics-14-01328-t001:** Physical characterization of the preliminary batches of nanocrystals prepared (using 10 mg of TZP) by homogenization (10 min) at 21,500 rpm with different stabilizers (**a**); and with PVA and POL (**b**).

Batches	Stabilizers (mg)/Concentrations (%, *w*/*v*)	Physical Characterization		
		Particle Size (nm)	Polydispersity Index	Zeta Potential (mV)
**(a) Preliminary trials of NCs * with 1.0 % (*w*/*v*) stabilizers**				
NC-P188	Polxamer-188	495.3 ± 23.5	0.265 ± 0.011	−16.2 ± 2.8
NC-PVA	PVA	514.8 ± 29.4	0.375 ± 0.023	−6.8 ± 1.9
NC-P407	Poloxamer-407 (F127)	614.6 ± 72.5	0.419 ± 0.012	−14.1 ± 3.2
NC-PVP	Polyvinylpyrrolidone K_30_	596.6 ± 45.2	0.279 ± 0.005	+5.5 ± 2.1
NC-TPGS	Vit-E TPGS	675.2 ± 77.6	0.439 ± 0.018	+9.7 ± 2.6
NC-SLS	Sodium lauryl sulfate	579.5 ± 36.7	0.512 ± 0.029	−1.2 ± 0.8
NC-Tween	Tween-80	563.7 ± 43.6	0.468 ± 0.008	−8.4 ± 2.4
**(b) Optimization of NCs * prepared with POL * or PVA * as stabilizers (Mean ± SD, *n* = 3)**				
NC-POL1	100 (0.5%, *w*/*v*)	715.2 ± 61.7	0.458 ± 0.024	−6.6 ± 3.5
NC-POL2	200 (1.0%, *w*/*v*)	481.7 ± 58.4	0.252 ± 0.021	−17.2 ± 5.6
NC-POL3	400 (2.0%, *w*/*v*)	603.0 ± 70.4	0.396 ± 0.006	−18.6 ± 4.0
NC-POL4	500 (2.5%, *w*/*v*)	600.5 ± 65.2	0.442 ± 0.009	−19.9 ± 6.9
NC-PVA1	100 (0.5%, *w*/*v*)	745.2 ± 89.1	0.403 ± 0.027	−0.6 ± 2.4
NC-PVA2	200 (1.0%, *w*/*v*)	613.8 ± 65.7	0.375 ± 0.019	−7.6 ± 3.4
NC-PVA3	400 (2.0%, *w*/*v*)	556.6 ± 59.5	0.328 ± 0.023	−6.3 ± 4.2
NC-PVA4	500 (2.5%, *w*/*v*)	507.6 ± 45.9	0.292 ± 0.007	−8.8 ± 3.9

* TZP = Tedizolid phosphate, NCs = Nanocrystals, PVA = Polyvinyl alcohol and POL = Poloxamer-188.

**Table 2 pharmaceutics-14-01328-t002:** Physical characterization of the NCs prepared using P188 at 1%, *w*/*v* (200 mg in each case) with varying durations of homogenization and probe sonication (Mean ± SD, *n* = 3) without mannitol.

Nanocrystals (NCs)	TZP (mg)	Homogenization Time (min) at 21,500 rpm	Sonication Time (s); 10 s Each Cycle	Physical Characterization (Mean ± SD, *n* = 3)			
				Particle Size (nm)	Polydispersity Index	Zeta Potential (mV)	TZP Content (%)
NCs-1	10	5	40	565.3 ± 42.7	0.428 ± 0.073	−5.8 ± 2.3	90.9 ± 1.5
NCs-2	10	7.5	50	389.5 ± 58.4	0.401 ± 0.021	−6.7 ± 3.9	90.1 ± 2.6
NCs-3	10	10	60	150.4 ± 18.3	0.231 ± 0.006	−13.5 ± 1.5	96.7 ± 1.2
NCs-4	10	15	70	147.5 ± 11.5	0.249 ± 0.012	−10.6 ± 9.9	92.4 ± 2.8

**Table 3 pharmaceutics-14-01328-t003:** Physical characterization and drug content of the different preparations of NCs-3 (TZP 10 mg and P188 at 1%, *w*/*v*) with Benzalkonium chloride (0.01%, *w*/*v*), stearylamine (0.2%, *w*/*v* of total formulation) and varying concentrations of mannitol (Mean ± SD, *n* = 3).

Nanocrystals (NCs)	Particle Size (nm)		PDI		Zeta Potential (mV)		TZP (%) Contents	
	Freeze Drying		Freeze Drying		Freeze Drying		Freeze Drying	
	Before	After	Before	After	Before	After	Before	After
NC_0 (No mannitol)_	150.4 ± 18.3	153.7 ± 16.6	0.231 ± 0.006	0.237 ± 0.003	+26.3 ± 5.1	+28.07 ± 5.0	93.1 ± 2.8	93.3 ± 2.8
NC_1 (1%, *w*/*v* mannitol)_	151.6 ± 17.5	154.3 ± 17.9	0.237 ± 0.005	0.243 ± 0.009	+29.4 ± 3.9	+31.64 ± 3.8	96.2 ± 2.5	96.4 ± 2.5
NC_2.5 (2.5%, *w*/*v* mannitol)_	157.5 ± 19.4	161.1 ± 19.3	0.344 ± 0.012	0.351 ± 0.016	+27.5 ± 5.6	+29.21 ± 5.6	92.4 ± 3.0	92.6 ± 2.9
NC_5 (5%, *w*/*v* mannitol)_	160.8 ± 18.7	163.2 ± 20.7	0.358 ± 0.016	0.365 ± 0.019	+28.9 ± 5.2	+30.87 ± 5.2	91.2 ± 2.6	91.6 ± 2.1

**Table 4 pharmaceutics-14-01328-t004:** Physicochemical characterization of redispersed NC_1_ and conventional TZP-AqS (mean ± SD, *n* = 3).

Formulations	Clarity at 25 °C	pH	Osmolarity (mOsm·L^−1^)	Drug Content (%)	Viscosity (cPs) at	
					20 ± 1 °C	35 ± 1 °C
* TZP-NC_1_ _(1%, *w*/*v* mannitol)_	Clear and transparent	7.0 ± 0.4	298.0 ± 5.0	96.4 ± 2.6	28.5 ± 1.2	21.1 ± 1.1
** TZP-AqS	Cloudy and translucent	6.2 ± 0.5	304.0 ± 4.0	98.4 ± 1.8	29.4 ± 2.1	23.1 ± 1.8

* TZP-NC_1_ = Tedizolid phosphate nanocrystals freeze-dried with 1%, *w*/*v* of mannitol) and ** TZP-AqS = Conventional aqueous suspension of tedizolid phosphate prepared in-house.

**Table 5 pharmaceutics-14-01328-t005:** Obtained parameters after applying the release kinetic models.

Release Models	Kinetic Parameters	TZP-NC_1_-AqS	Conventional TZP-AqS
Zero order (Fraction drug released vs. time)	*R* ^2^	0.9157	0.8309
	*n*-value	0.2631	0.0138
	*k*_0_ (µgh^−1^)	1.11 × 10^−1^	0.53 × 10^−1^
First order (Log% Drug remaining vs. time)	*R* ^2^	0.9875	0.9728
	*n*-value	0.0238	0.0081
	*k*_1_ (h^−1^)	2.28	2.37 × 10^−1^
Korsmeyer-Peppas (Log Fraction drug released vs. log time)	*R* ^2^	0.9685	0.9645
	*n*-value	0.3221	0.3534
	*k_K-P_* (h^−n^)	4.52 × 10^−1^	2.39 × 10^−1^
Hixon-Crowell (M_o_^1/3^–M_t_^1/3^ vs. time)	*R* ^2^	0.9699	0.8386
	*n*-value	0.0141	0.0054
	*k_H-C_* (µg^1/3^h^−1^)	4.79 × 10^−1^	4.77 × 10^−1^

*R*^2^ = Coefficient of correlation and *n*-value = Release or diffusion exponent.

**Table 6 pharmaceutics-14-01328-t006:** Stability results of TZP-NC_1_. All data were presented as mean ± SD, *n* = 3.

Stability of TZP-NC_1_		Values at Different Time Points (Mean ± SD, *n* = 3)			
	Initially (0 Day)	At 7 Days	At 30 Days	At 90 Days	At 180 Days
At 4 ± 2 °C					
Particle size (nm)	154.3 ± 17.9	157.2 ± 14.7	159.8 ± 15.2	161.8 ± 15.6	164.1 ± 15.8
Polydispersity index	0.242 ± 0.009	0.244 ± 0.008	0.245 ± 0.008	0.247 ± 0.009	0.249 ± 0.011
Zeta potentials (mV)	+31.6 ± 3.8	+31.4 ± 3.7	+30.8 ± 3.8	+30.2 ± 3.7	+29.4 ± 3.5
Drug content (%)	96.2 ± 3.1	96.17 ± 2.56	95.8 ± 2.8	95.2 ± 3.4	94.5 ± 3.6
At 25 ± 1 °C					
Particle size (nm)	154.3 ± 17.9	159.1 ± 14.9	161.6 ± 15.1	174.2 ± 16.4	190.0 ± 17.8
Polydispersity index	0.242 ± 0.009	0.247 ± 0.008	0.248 ± 0.008	0.251 ± 0.009	0.253 ± 0.008
Zeta potentials (mV)	+31.6 ± 3.8	+31.0 ± 3.7	+30.6 ± 3.6	+29.4 ± 3.4	+29.1 ± 3.4
Drug content (%)	96.2 ± 3.1	96.1 ± 2.8	95.6 ± 3.2	95.4 ± 3.5	94.9 ± 3.4
At 37 ± 1°C					
Particle size (nm)	154.3 ± 17.9	161.0 ± 15.2	162.8 ± 15.3	176.4 ± 16.6	192.1 ± 18.0
Polydispersity index	0.242 ± 0.009	0.247 ± 0.007	0.249 ± 0.008	0.252 ± 0.009	0.255 ± 0.008
Zeta potentials (mV)	+31.6 ± 3.8	+30.6 ± 3.8	+30.3 ± 3.6	+29.0 ± 3.4	+28.6 ± 3.4
Drug content (%)	96.2 ± 3.1	95.8 ± 2.9	95.6 ± 3.1	95.3 ± 3.5	94.2 ± 3.2

## Data Availability

This study did not report any data related animal study.

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
