# Peer review of "Fabrication and Characterization of Tedizolid Phosphate Nanocrystals for Topical Ocular Application: Improved Solubilization and In Vitro Drug Release"

_pharmaceutics, 2022, doi:10.3390/pharmaceutics14071328_

Round 1

Reviewer 1 Report

The manuscript presented by Kalam et al., deals with the development, characterization and in vitro evaluation of Tedizolid nanocrystals for ocular drug delivery. The manuscript is interesting and the characterization was well conducted.

In order to contribute/improve the manuscript, the authors should observe the following:

Introduction. First paragraph. The authors cannot generalize that conventional ocular formulations are not effective for therapeutics. They have limitations and disadvantages and this can be an issue for some APIs.

Pg4. Second paragraph. Ref 32 is too old to support the information that patent is a limiting factor for this kind of formulation to be marketed.

2.7. Please clarify the temperature used in solubility studies. It was 37oc in shaker, but after 72hs it was left for 24hs to settle solid particles. In this 24hs period, was the temperature controlled? Why not centrifuging after 72hs? It is the common procedure.

2.7. Please detail the simulated tear fluid and include a reference.

3.1. It is not necessary and relevant here. It could be more detailed and include as supplementary material.

3.2 define HLB.

3.6 Thermograms is not the term recommended by IUPAC/ICTAT. Please change for curve.

2.8 and 3.11. The f1 and f2 parameters are not the most adequate analysis here. The profiles are very distinct. Maybe it could be substituted by Dissolution efficiency with statistical analysis.

2.9. Include a reference and/or a justification for the temperatures chosen. Usually a 40oC is evaluated instead of 37oC.

Author Response

Reviewer #1: Comments and Suggestions for Authors

The manuscript presented by Kalam et al., deals with the development, characterization and in vitro evaluation of Tedizolid nanocrystals for ocular drug delivery. The manuscript is interesting and the characterization was well conducted. In order to contribute/improve the manuscript, the authors should observe the following:

Response: We thank for the valuable comments by the reviewer on our manuscript. The comments and suggestions have greatly improved the quality of the manuscript. We have tried our best to accommodate the recommendations of the learned reviewer in the revised version of this manuscript. All the modifications/changes are highlighted as yellow text in the revised manuscript.

Introduction. First paragraph. The authors cannot generalize that conventional ocular formulations are not effective for therapeutics. They have limitations and disadvantages and this can be an issue for some APIs.

Response: As per the suggestion, the statement has been modified and included in the manuscript. “Some of the conventional formulations for ODD including the eye drops as solutions, suspensions, emulsions etc. containing poorly soluble or low permeable drugs have shown low ocular availability (approximately 1-5% of the applied dose) of the drugs 

Pg4. Second paragraph. Ref 32 is too old to support the information that patent is a limiting factor for this kind of formulation to be marketed.

Response: Thanks for the valuable advice. We have modified the statement considering the fact about the future of NCs in drug delivery applications and cited one new reference also. “The NCs can be a successful alternative to the pharmaceutical industries and have commercial victory in oral, parenteral and ODD of highly lipophilic, poorly soluble and low permeable drugs [1, 2]. We believe that in near future the NCs would support the development of robust and effective therapies for ocular disorders”.

[1] Chen, Z.; Wu, W.; Lu, Y. What is the future for nanocrystal-based drug-delivery systems? Therapeutic Delivery 2020, 11, 225-229.

[2] Muller, R.H.; Keck, C.M. Twenty years of drug nanocrystals: where are we, and where do we go? Eur J Pharm Biopharm 2012, 80, 1-3, doi:10.1016/j.ejpb.2011.09.012.

2.7. Please clarify the temperature used in solubility studies. It was 37ºC in shaker, but after 72hs it was left for 24hs to settle solid particles. In this 24hs period, was the temperature controlled? Why not centrifuging after 72hs? It is the common procedure.

Response: Thanks for the reviewer for reminding about the procedure. The procedure was followed as mentioned by the reviewer but the method was not properly mentioned in the methodology section. The method has been updated now Section: Solubility of Tedizolid phosphate (TZP) of the revised manuscript. 

2.7. Please detail the simulated tear fluid and include a reference.

Response: The STF was obtained by dissolving 6.8 g of NaCl, 2.2 g of NaHCO3, 1.4 g of KCl and 0.08 g of CaCl2.2H2O in 1000 mL of purified water to get the STF by following two mentioned references [3, 4] which were included in Section: Solubility of Tedizolid phosphate (TZP) in revised manuscript

REF:

[3] Alkholief, M.; Kalam, M.A.; Almomen, A.; Alshememry, A.; Alshamsan. Thermoresponsive sol-gel improves ocular bioavailability of Dipivefrin hydrochloride and potentially reduces the elevated intraocular pressure in vivo. 2020, 28, 1019-1029.

[4] Kalam, M.A.; Iqbal, M.; Alshememry, A.; Alkholief, M.; Alshamsan, A. Development and Evaluation of Chitosan Nanoparticles for Ocular Delivery of Tedizolid Phosphate. Molecules 2022, 27, 2326.

3.1. It is not necessary and relevant here. It could be more detailed and include as supplementary material.

Response: The section 3.1 has been included in the Supplementary materials.

3.2 define HLB.

Response: It is now mentioned as separate section “Abbreviations”. The HLB stand for Hydrophilic-Lipophilic Balance, signifies the surface acting or stabilizing properties of any surfactant or stabilizer.

3.6 Thermograms is not the term recommended by IUPAC/ICTAT. Please change for curve.

Response: As per the suggestion, “thermograms” was changed to “curves” in the revised manuscript

 2.8 and 3.11. The f1 and f2 parameters are not the most adequate analysis here. The profiles are very distinct. Maybe it could be substituted by Dissolution efficiency with statistical analysis.

Response: As per the suggestion we tried our best to substitute the fit factors (f1 and f2) with the Dissolution Efficiency more appropriate and quantitative comparison of in vitro dissolution profiles between batches

2.9. Include a reference and/or a justification for the temperatures chosen. Usually a 40ºC is evaluated instead of 37ºC.

Response: Thanks for the suggestion. It is true that the stability study should be performed at little higher temperature (such as 40ºC), although some researchers and our group have performed the stability study of nanoparticles at 37ºC (REF. 5-8). Also the stability of elastin-PLGA-NPs was at 37°C for up to 30 days was done by Stromberg et al., 2021 [9]. Therefore, we followed the same trend in this work also. In our future studies we must follow the reviewer’s valuable suggestion to perform the stability study of nano-formulations at 40ºC.

REFERENCES:

[5] The stability of Chitosan-coated poly (lactic-co-glycolide) nanoparticles was performed at 30 ± 1°C. [Abdullah Alshememry, Mohd Abul Kalam, Abdulhadi Almoghrabi, Abdulwahab Alzahrani, Mudassar Shahid, Azmat Ali Khan, Anzarul Haque, Raisuddin Ali, Musaed Alkholief, Ziyad Binkhathlan, Aws Alshamsan. Chitosan-coated poly (lactic-co-glycolide) nanoparticles for dual delivery of doxorubicin and naringin against MCF-7 cells. Journal of Drug Delivery Science and Technology 68 (2022), 103036

[6] Similarly, the stability on Tacrolimus-loaded PLGA-NPs was performed and stored for 30 days at 25±1°C for 30 days only. [Mohd Abul Kalam, Aws Alshamsan. Poly (D, L-lactide-co-glycolide) nanoparticles for sustained release of tacrolimus in rabbit eyes. Biomedicine & Pharmacotherapy 94 (2017) 402–411].

[7] The stability study of atorvastatin calcium loaded PLGA-NPs was evaluated at 37 °C for 10 days. [Z. Li, W. Tao, D. Zhang, C. Wu, B. Song, S. Wang, T. Wang, M. Hu, X. Liu, Y. Wang, The studies of PLGA nanoparticles loading atorvastatin calcium for oral administration in vitro and in vivo, Asian J. Pharm. Sci. 12 (3) (2017) 285–291].

[8] Anita Hafner, Jasmina Lovrić, Dario Voinovich, Jelena Filipović-Grčić. Melatonin-loaded lecithin/chitosan nanoparticles: Physicochemical characterisation and permeability through Caco-2 cell monolayers. Int. J Pharm. 381 (2), 2009; 205-213.

[9] Zachary R. Stromberg, M.Lisa Phipps, Harsha D. Magurudeniy, Christine A. Pedersen, Trideep Rajale, Chris J. Sheehan, Samantha J. Courtney, Steven B. Bradfute, Peter Hraber, Matthew N. Rush, Jessica Z. Kubicek-Sutherland, Jennifer S. Martinez. Formulation of stabilizer-free, nontoxic PLGA and elastin-PLGA nanoparticle delivery systems. Int. J Pharm. 597, 2021, Article # 120340.

Reviewer 2 Report

Authors proposed a paper entitled “Fabrication and Characterization of Tedizolid Phosphate Nanocrystals for Topical Ocular Application: Improved Solubilization and In vitro Drug Release” for the publication in Pharmaceutics, mdpi.

This paper has a good scientific soundness and the overall use of English is quite good.

I suggest adding an abbreviation list, according to the guidelines of this Journal.

Here is the list of some issues:

Abstract

Line 2. “was prepared” should be plural: were prepared.

“Mannitol (1%)”, is by weight intended? Please add w/w % to percentages expressed in this paper, in terms of mass basis. For example, I have noticed that in some cases the percentage is, for example, 5 % (w/v). Therefore, please specify the unit of measure for these percentages.

Introduction

I would suggest the addition of a higher number of support references. An example could be after the first sentence “Due to unique anatomy and physiology of eyes, the development of ocular drug delivery (ODD) is a major challenge to the formulators.”

“Nanotechnology-based ODD approaches” I would also talk about conventional and high pressure production processes of drug carriers. Of particular importance, supercritical assisted processes could be described.

“Still, the application of NCs has not been explored well for ocular applications” I would check the use of English here.

“the majority of production technologies for NCs are patented” however, what about the patens of products?

The suspensions of NCs of many poorly soluble drugs such as dexamethasone, hydrocortisone and prednisolone have been explored to improve the ocular bioavailability as compared to their micro-crystalline suspensions and aqueous solutions.” This sentence should be supported by references.

Materials. May you add CAS numbers for the materials employed in your work?

“251 nm” do authors mean the wavelength set for this molecule detection? Please be more specific.

Paragraph 2.8. “so the release pattern”; I will substitute the more informal “so” with “therefore…”

Stabiliy study in paragraph 2.9. Please check also this paper as a comparison: “ https://www.sciencedirect.com/science/article/pii/S0896844618306156 ”

Table 1. Please, use “plus” symbol for positive zeta potential values.

Particle sizes in Table 1 should be defined as mean values plus/minus standard deviations, also for preliminary trials.  

Table 2. TZP value could be eliminated from the table and inserted in the caption as a set variable.

Please describe the problem of solvent elimination from the drug formulations, as partially reported in page 16. Add references on supercritical assisted processes of extraction.

“Here, we applied 60 sec probe sonication during the preparation process” informal expressions and description that should be moved to methods section.

Pag. 18. “dispersion ability when come in contact with aqueous phase” I would replace with “in case of contact with aqueous phase”.

Paragraph 3.3 should contain size and zeta potential values in terms of mean size plus/minus standard deviations.

Paragraph 3.4. The value “565.3 nm” comes from an average of more than one measurement on the same sample; I would suggest an approximation to zero decimals.

Figure Nr. 3 is a screenshot from Zetasizer instrument report. I would suggest extracting raw data and re-draw them using a drawing program such as Origin or several others. Moreover I would also show the particle size distributions of the other samples, maybe on the same comparison diagram.

Figure Nr. 4 is characterized by 5 sub-diagrams. However, to help the reader, it is necessary to add “a”, “b” to “e” next to each diagram of this group of figures, instead of occupying the internal area of the diagram.

Same observation for the composite figure Nr. 5.

I would not start the section 3.11 with the drug release diagram. I would first introduce it. Perhaps it would be better to re-draw this diagram by setting y-axis to 100%. Please, discuss the advantage of increasing the release kinetics in this study compared with conventional formulation.

Repetitions of "revealed" in the conclusion section.

Please, check the format of the references, according to the guidelines of this Journal.

Author Response

Reviewer # 2: Comments and Suggestions for Authors

Authors proposed a paper entitled “Fabrication and Characterization of Tedizolid Phosphate Nanocrystals for Topical Ocular Application: Improved Solubilization and In vitro Drug Release” for the publication in Pharmaceutics, MDPI. This paper has a good scientific soundness and the overall use of English is quite good.

Response: We thank to the reviewer for the positive and valuable comments on our manuscript. The comments and suggestions have greatly improved the quality of the manuscript.

I suggest adding an abbreviation list, according to the guidelines of this Journal.

Response: The Abbreviation list has been added in the manuscript

Here is the list of some issues: 

Abstract

Line 2. “was prepared” should be plural: were prepared.

Response: Corrected as suggested

“Mannitol (1%)”, is by weight intended? Please add w/w % to percentages expressed in this paper, in terms of mass basis. For example, I have noticed that in some cases the percentage is, for example, 5 % (w/v). Therefore, please specify the unit of measure for these percentages.

Response: As suggested, at all places the unit of strength was corrected as (%, w/v)

Introduction

I would suggest the addition of a higher number of support references. An example could be after the first sentence “Due to unique anatomy and physiology of eyes, the development of ocular drug delivery (ODD) is a major challenge to the formulators.”

Response: The references have been added at the specified places in the revised version of the manuscript

“Nanotechnology-based ODD approaches” I would also talk about conventional and high pressure production processes of drug carriers. Of particular importance, supercritical assisted processes could be described.

Response: As per suggestion supercritical assisted techniques for nan-formulation development, top-down or bottom-up methods for nanocrystals have been included in the revised manuscript.

“Still, the application of NCs has not been explored well for ocular applications” I would check the use of English here.

Response: The sentence has been corrected “The application of NCs has not been well explored for ocular applications”.

“the majority of production technologies for NCs are patented” however, what about the patens of products?

Response: The sentence has been modified and few references have been included regarding the product patents of nanocrystals for drug delivery.

“The suspensions of NCs of many poorly soluble drugs such as dexamethasone, hydrocortisone and prednisolone have been explored to improve the ocular bioavailability as compared to their micro-crystalline suspensions and aqueous solutions.” This sentence should be supported by references.

Response: The references have been included to support this statement

Materials. May you add CAS numbers for the materials employed in your work?

Response: The CAS of the materials have been included in the Materials section of the revised manuscript.

 “251 nm” do authors mean the wavelength set for this molecule detection? Please be more specific.

Response: Yes, the elute was detected at 251 nm wavelength, specified as per suggestion.

Paragraph 2.8. “so the release pattern”; I will substitute the more informal “so” with “therefore…”

Response: “so” has been replaced with “therefore”

Stabiliy study in paragraph 2.9. Please check also this paper as a comparison: “ https://www.sciencedirect.com/science/article/pii/S0896844618306156 ”

Response: The suggested article has been cited in the methods for stability study. The results obtained were discussed by comparing with the given reference.

Table 1. Please, use “plus” symbol for positive zeta potential values.

Response: The “plus” sign for the zeta potentials has been included now.

Particle sizes in Table 1 should be defined as mean values plus/minus standard deviations, also for preliminary trials.  

Response: All the values mentioned in Table 1 have been presented as mean with SD, also for the preliminary trials.  

Table 2. TZP value could be eliminated from the table and inserted in the caption as a set variable.

Response: The value for TZP has been removed from the Table 2, and put in the caption.

Please describe the problem of solvent elimination from the drug formulations, as partially reported in page 16. Add references on supercritical assisted processes of extraction.

Response: As per the suggestion following information has been updated in the manuscript with proper references. The organic solvents were then evaporated by continuous magnetic stirring (12 h) at room temperature to get the suspension of NCs. The supercritical fluid process (SFC) has advantage over this process in terms of rapid removal of fluids and solvents without requiring the extensive drying steps as compared to other solvent precipitation methods. It works with supercritical carbon dioxide (SCO2, at temperature ≈31.1ºC and pressure 73.8 Bar) for most of the pharmaceuticals. Due to low polarity of SCO2, the solubilization of lipophilic drugs are easy to form the solution and passing of the drug solution through the capillary tube into an ambient surroundings forms the fine/small particles.

“Here, we applied 60 sec probe sonication during the preparation process” informal expressions and description that should be moved to methods section.

Response: As per the suggestion, the sentence has been shifted into method section as “The probe sonication during this process provided added stability to the TZP-NCs by minimizing the metastable zone for TZP and its super-saturation level”.

Pag. 18. “dispersion ability when come in contact with aqueous phase” I would replace with “in case of contact with aqueous phase”.

Response: Corrected as suggested.

Paragraph 3.3 should contain size and zeta potential values in terms of mean size plus/minus standard deviations.

Response: The size, polydispersity and zeta potential values were mentioned as mean with SD, in this section. The section 3.3 will be section 3.2 in the revised manuscript, because one of the reviewer suggested to shift the section 3.1 as Supplementary materials.

Paragraph 3.4. The value “565.3 nm” comes from an average of more than one measurement on the same sample; I would suggest an approximation to zero decimals.

Response: Corrected as suggested, now this section can be read as section 3.3.

Figure Nr. 3 is a screenshot from Zetasizer instrument report. I would suggest extracting raw data and re-draw them using a drawing program such as Origin or several others. Moreover I would also show the particle size distributions of the other samples, maybe on the same comparison diagram.

Response: We highly appreciate the reviewer’s suggestion about the redrawing of the distribution curves. We should follow this method to redraw the curves, but these would be little hectic at this time. Therefore, we must consider this procedure in our future studies. Presently we are providing the particle size distribution curves of other samples also for comparative analysis as per the reviewers’ advice.

Figure Nr. 4 is characterized by 5 sub-diagrams. However, to help the reader, it is necessary to add “a”, “b” to “e” next to each diagram of this group of figures, instead of occupying the internal area of the diagram.

Response: As per the suggestion, the name symbol of sub-diagrams have been shifted next to each sub-diagram of this group of Figures (now it is Figure 5).

Same observation for the composite figure Nr. 5.

Response: As per the suggestion, the sub-figures names as “a” to “e” have been shifted next to each diagram of this group of figures (now it is Figure 6)

I would not start the section 3.11 with the drug release diagram. I would first introduce it. Perhaps it would be better to re-draw this diagram by setting y-axis to 100%. Please, discuss the advantage of increasing the release kinetics in this study compared with conventional formulation.

Response: The re-drown Figure 7 (now Figure 8, In vitro release profile of TZP from the two formulations) has been included in the revised manuscript by setting the Y-axis to 100%.  Application of some kinetic models on the in vitro release data we found that the in vitro release of TZP from both the formulations followed the first order release kinetics. Although the release kinetic of the drug from the formulations was same, but the release of TZP was linear till 12 h from the NC1 while it was linear upto 8 h from the TZP-AqS, thereafter the drug release remained almost same till 12 h. Among the applied kinetic equations or release models the highest correlation coefficients (R2) were 0.9875 and 0.9645 (for TZP-NC1 and AqS of TZP-pure, respectively) (Figure S1), which were associated with the first order model (Table 5). By taking the R2 values and slope of the tried kinetic equations, the diffusion-exponents (n-value) were obtained. The n-values (0.0238 and 0.0078 for TZP-NC1 and AqS of TZP-pure, respectively) indicating the Fickian-Diffusion type of release mechanism.

Repetitions of "revealed" in the conclusion section.

Response: Corrected as suggested

 Please, check the format of the references, according to the guidelines of this Journal.

Response: Thanks for the suggestion, we have checked and corrected the references as per the journal requirement.

Reviewer 3 Report

The manuscript hasn’t been prepared using the MDPI template. Next time please do this as it really facilitates the review process, i.e. continuous line numbering is really helpful.

The language must be greatly improved, even in the very first sentence of an abstract there is a grammar mistake (“was” instead of “were”).

The abstract is significantly too long, it should be maximum 200 words, right now it is 381.

There are too many references (over 100) to the previously reported studies. This number must be decreased.

Why the Authors keep writing tedizolid using capital T? It also concerns other chemicals (i.e. Detrose).

The introduction part is very long too. On the other hand it lacks the figure with chemical structure of TDZ. Further, in the introduction the Authors should state why measuring Zeta potential is so important in this case.

The Methods Part (2.2.) is very well prepared, good job!

Tables (all of them), the uncertainties are not rounded properly. The probable uncertainty of the result should be rounded to one significant figure. The last significant figure in the final result itself should be of the same order of magnitude (in the same decimal position) as the uncertainty. The probable error is always rounded upwards and uncertainty should not be more than 15 units (two significant digits) If the probable error should according to the rules 1 and 2 be rounded so that the non-zero number is equal to 2, but the actual value is closer to 1, the uncertainty is given with two significant figures.

Page 21, “ as the amorphous compound does not exhibit melting point (endothermic peaks).” What does it exactly mean? That the amorphous compound never melts? How is it possible? Please record the DSC of the amorphous compound. Otherwise, those thesis are not credible.

Page 22, “he decreased lattice energy”-you don’t measure the lattice energy by the DSC. Those are enthalpies.

Figure 5, please remove the BRUKER logo to increase the clarity.

From the PXRD measurements the Authors could easily determine the unit cell dimensions and crystal size using the Quantitative Phase Analysis method and one of many available free software. Why the Authors haven’t tried that?

Page 33, “The calculated values of f1 and f2 for TZP-NC1 in the present release media were found to be 85.3 (greater than 15) and 30.8 (less than 50), respectively.”-so what does this mean? Why “greater than 15” and “less than 50” is so important?

At the end, the part describing each Author’s individual contribution is missing. This is mandatory in Pharmaceutics.

Author Response

Reviewer #3: Comments and Suggestions for Authors

The manuscript hasn’t been prepared using the MDPI template. Next time please do this as it really facilitates the review process, i.e. continuous line numbering is really helpful.

Response: Thanks for the advice, we will keep in mind for our future submission.

The language must be greatly improved, even in the very first sentence of an abstract there is a grammar mistake (“was” instead of “were”).

Response: We tried our best to improve the English language of this manuscript.

The abstract is significantly too long, it should be maximum 200 words, right now it is 381.

Response: We tried hard to reduce the number of words as possible.

There are too many references (over 100) to the previously reported studies. This number must be decreased.

Response: We tried hard to reduce the number of references, but the another reviewer advice to add many new references in the introduction, methods, results and discussion.  So the total number of references are still very high.

Why the Authors keep writing tedizolid using capital T? It also concerns other chemicals (i.e. Dextrose).

Response: We have considered the point raised by the reviewer and through the manuscript we presented as “tedizolid”, and “dextrose” which was also adopted for other chemicals also.

The introduction part is very long too. On the other hand it lacks the figure with chemical structure of TDZ. Further, in the introduction the Authors should state why measuring Zeta potential is so important in this case.

Response: The introduction part was tried to shorten. The chemical structure of TDZ was included now. The importance of zeta-potential for the products to be used as ocular drug delivery has been also incorporated in the introduction part.

The Methods Part (2.2.) is very well prepared, good job!

Response: We thank the reviewer for the appreciation

Tables (all of them), the uncertainties are not rounded properly. The probable uncertainty of the result should be rounded to one significant figure. The last significant figure in the final result itself should be of the same order of magnitude (in the same decimal position) as the uncertainty. The probable error is always rounded upwards and uncertainty should not be more than 15 units (two significant digits) If the probable error should according to the rules 1 and 2 be rounded so that the non-zero number is equal to 2, but the actual value is closer to 1, the uncertainty is given with two significant figures.

Response: All the Tables have been revised well and the values/ data were mentioned up to 1 digit after decimal throughout the manuscript for most of the parameters except for the PDI; which were in three decimal places, because the lower most value (< 1) of PDI is considered as best

Page 21, “as the amorphous compound does not exhibit melting point (endothermic peaks).” What does it exactly mean? That the amorphous compound never melts? How is it possible? Please record the DSC of the amorphous compound. Otherwise, those thesis are not credible.

Response: As per the comment of the expert reviewer, we have modified these sentences in the revised manuscript.

Page 22, “he decreased lattice energy”-you don’t measure the lattice energy by the DSC. Those are enthalpies.

Response: The sentence has been corrected in the revised manuscript

Figure 5, please remove the BRUKER logo to increase the clarity.

Response: Thanks for the advice, we tried our best to remove the logo

From the PXRD measurements the Authors could easily determine the unit cell dimensions and crystal size using the Quantitative Phase Analysis method and one of many available free software. Why the Authors haven’t tried that?

Response: As per the suggestion of the learned reviewer, we tried our best to utilize the data obtained from the XRD analysis to calculate the crystallite size, applying the Scherrer equation as mentioned in the methods of XRD study. The results so obtained were mentioned in the discussion part of XRD analysis.   

Page 33, “The calculated values of f1 and f2 for TZP-NC1 in the present release media were found to be 85.3 (greater than 15) and 30.8 (less than 50), respectively.”-so what does this mean? Why “greater than 15” and “less than 50” is so important?

Response: The calculation of difference factor (f1) and Similarity factor (f2) are the model independent mathematical approach compares the release profiles of two different products [1]. Where, f1 is directly proportional to the average difference between the release profiles and f2 is inversely proportional to the average squared difference of the release profiles of drug from two different products [2]. An f1 value lower than 15 and f2 value of 50 or higher than 50 to 100 represents equivalence of the two release curves and therefore, the performance of the two drug products if they differ from each other it means the two products have different release profiles [3]. Therefore, “greater than 15” and “less than 50” was considered in the present investigation.

As per the suggestion other reviewer the f1 and f2 are very distinct, may it could be substituted by Dissolution efficiency with statistical analysis” [4, 5]. Therefore, these have been substituted by the calculation of dissolution efficiency (%DE) with statistical analysis and updated in the revised manuscript.

[1] Moore, J.W.; Flanner, H.H. Mathematical comparison of dissolution profiles. Pharmaceutical technology 1996, 20, 64-74.

[2] Shah, V.P.; Tsong, Y.; Sathe, P.; Liu, J.P. In vitro dissolution profile comparison--statistics and analysis of the similarity factor, f2. Pharm Res 1998, 15, 889-896, doi:10.1023/a:1011976615750.

[3] Samaha, D.; Shehayeb, R.; Kyriacos, S.J.D.T. Modeling and comparison of dissolution profiles of diltiazem modified-release formulations. 2009, 16, 41-46.

[4] Kassaye, L.; Genete, G. Evaluation and comparison of in-vitro dissolution profiles for different brands of amoxicillin capsules. Afr Health Sci 2013, 13, 369-375, doi:10.4314/ahs.v13i2.25.

[5] Anderson, N.H.; Bauer, M.; Boussac, N.; Khan-Malek, R.; Munden, P.; Sardaro, M. An evaluation of fit factors and dissolution efficiency for the comparison of in vitro dissolution profiles. J Pharm Biomed Anal 1998, 17, 811-822, doi:10.1016/s0731-7085(98)00011-9.

At the end, the part describing each Author’s individual contribution is missing. This is mandatory in Pharmaceutics.

Response: The authors individual contributions have been included at the end of the manuscript as per the guideline of this journal.

Round 2

Reviewer 1 Report

The authors have accepted all the suggestions.

I have only one minor comment. 

1- The authors should include the release kinetic parameters calculated with the respective reference in the Section 2.8 In vitro release study OR detail it in supplementary material.

Author Response

Comments and Suggestions for Authors by Reviewer #1

The authors have accepted all the suggestions.

I have only one minor comment.

  • The authors should include the release kinetic parameters calculated with the respective reference in the Section 2.8 In vitro release study OR detail it in supplementary material.

Response: We thanks reviewer for this valuable comment. We have accommodated the fruit suggestion by the esteemed reviewer. For the determination of release kinetics and mechanism of drug release, the in vitro release data were fitted into different kinetic models such as the zero-order, first-order, Korsmeyer-Peppas and Hixson-Crowell models as mentioned in method sub-section 2.8. The values of release kinetic parameters for the two formulations after applying the kinetic models, were mentioned in Table 5. These are represented in the results and discussion part in sub-section: 3.10 In vitro release study.

Reviewer 2 Report

Authors provided a revised version of their manuscript.

Although the revisions have improved the level of the paper, a revision of the use of English should be again considered.

An example is “… have commercial victory”. May the author really use in this context such expression? Please check it.

“73.8 Bar”. Bar should not be in capital letter.

Fig. 4. It would be better to extract raw data from the instrument output/report and draw particle size distributions.

Table 4.Please, correct mOsmolL-1

Figure 8. Define clearly, whether authors considered encapsulation efficiency in drug release diagram.

Table 5. Indicate the number of days, not first days and then months.

I would add these two citations about ocular delivery using high pressure CO2 assisted systems.

Yañez, F., Martikainen, L., Braga, M. E., Alvarez-Lorenzo, C., Concheiro, A., Duarte, C. M., ... & De Sousa, H. C. (2011). Supercritical fluid-assisted preparation of imprinted contact lenses for drug delivery. Acta biomaterialia, 7(3), 1019-1030.

Campardelli, R., Trucillo, P., & Reverchon, E. (2018). Supercritical assisted process for the efficient production of liposomes containing antibiotics for ocular delivery. Journal of CO2 Utilization, 25, 235-241.

Author Response

Comments and Suggestions for Authors by Reviewer #2

Authors provided a revised version of their manuscript.

Response: We thank for the valuable comments by the reviewer. We have tried our best to include the comments and recommendations. These considerations have greatly improved the quality of the manuscript now. All the modifications/changes are highlighted as green texts in the revised manuscript.

Although the revisions have improved the level of the paper, a revision of the use of English should be again considered.

An example is “… have commercial victory”. May the author really use in this context such expression? Please check it.

Response: Thanks for this valuable suggestion. The sentences have been corrected as per the instructions and a thorough revision of the manuscript was performed to improve the English 

“73.8 Bar”. Bar should not be in capital letter.

Response: Thanks for the suggestion, it is corrected now as “bar”.

Fig. 4. It would be better to extract raw data from the instrument output/report and draw particle size distributions.

Response: Thanks for the reviewer’s suggestion. We have extracted the raw data from the Zetasizer and draw the particle size and zeta-potential distribution curves using OriginPro 8.5 software, as represented in Figure 4.

Table 4. Please, correct mOsmolL-1

Response: Thanks for the reviewer’s suggestion. As per the suggestion, the unit of osmolarity has been corrected as mOsm.L-1 at places throughout the manuscript

Figure 8. Define clearly, whether authors considered encapsulation efficiency in drug release diagram.

Response: Thanks for the reviewer’s valid point. The drug release study was performed by taking an equivalent amount of NCs containing TZP (0.1%, w/v) by considering the “drug content”, which was included in the method of in vitro release study.

Table 5. Indicate the number of days, not first days and then months.

Response: Thanks for the reviewer’s suggestion. Now it has been corrected as number of days rather than months

I would add these two citations about ocular delivery using high pressure CO2 assisted systems.

Yañez, F., Martikainen, L., Braga, M. E., Alvarez-Lorenzo, C., Concheiro, A., Duarte, C. M., ... & De Sousa, H. C. (2011). Supercritical fluid-assisted preparation of imprinted contact lenses for drug delivery. Acta biomaterialia7(3), 1019-1030.

Campardelli, R., Trucillo, P., & Reverchon, E. (2018). Supercritical assisted process for the efficient production of liposomes containing antibiotics for ocular delivery. Journal of CO2 Utilization25, 235-241.

Response: Thanks for the reviewer’s suggestion. We have included these two citations [Ref # 17 and 59 in the manuscript] about ocular deliveries of flurbiprofen through soft contact lens as well as ampicillin and ofloxacin from the supercritical assisted liposomes which were developed by high pressure CO2 assisted techniques.

Reviewer 3 Report

The Authors did their best to improve the manuscript. However, the quality of figure 6 (spectra) is still very low. the Authors have tried to remove the Bruker logo, however it has not been done properly. Please contact the person who has recorded those spectra and ask for a result without this logo. If it is not possible, the previous version of this Figure was better (even with this unwanted logo).

Author Response

Comments and Suggestions for Authors by Reviewer #3

The Authors did their best to improve the manuscript. However, the quality of figure 6 (spectra) is still very low. the Authors have tried to remove the Bruker logo; however, it has not been done properly. Please contact the person who has recorded those spectra and ask for a result without this logo. If it is not possible, the previous version of this Figure was better (even with this unwanted logo).

Response: Thanks for the valuable suggestion by the learned reviewer. We have contacted the person who has recorded the FTIR spectra, they could not provide the spectra all together without BRUKER logo. Therefore, we again tried to remove the logo in the same Figure as submitted earlier. Now it is looking better, as represented in Figure 6.

Round 3

Reviewer 2 Report

Authors provided a new version of the paper.

Now the research paper deserves to be published in the present version.

Thank you.